# Interfacial Ru/RuO$_x$ heterostructures on carbon support regulate selectivity in lignin hydrodeoxygenation

Hongfei Ma[1,2,9] ✉, Chen Chen[3,4,9], Guoyan Ma[5,9], Tian Ouyang[6], Hao Zhang [7], Qixian Wang[1], Wenjing Wei[5], Yifan Li[8], Yi Cui[8] ✉ & De Chen [2] ✉

Constructing well-defined heterostructure interfaces in catalysts provides an approach to modulate scaling constraints and steer reaction pathways in biomass upgrading. Herein, we demonstrate that thermal restructuring of hydroxyl groups on carbon nanofibers (CNF) induces the formation of heterostructures of Ru/RuO$_x$, which function as bifunctional active sites for the one-pot hydrodeoxygenation (HDO) of lignin to liquid hydrocarbons. The optimized 5 wt% Ru/CNF catalyst delivers promising performance, achieving a mass/carbon yield of 49.1%/67.7%, with high selectivity toward saturated cycloalkanes. X-ray absorption spectroscopy and near-ambient pressure X-ray photoelectron spectroscopy confirm that thermal treatment of CNF tunes the oxidation state of Ru. DFT calculations reveal that O-rich Ru/CNF forms interfacial heterostructures of Ru/RuO$_x$ polarized active sites, characterized by O$^{\delta-}$···Ru$^{\delta++}$···Ru$^{\delta+}$ ensembles that heterolytically activate H$_2$ and strongly polarize C-O bonds in phenolic intermediates. The cooperative interplay between metallic Ru and partially oxidized RuO$_x$ interfacial sites lowers the energy barriers for hydrogenation and deoxygenation reactions, enabling a cooperative reaction pathway. These insights elucidate the molecular basis of tunable selectivity in lignin HDO and demonstrate that a polarized, oxygen-decorated metal-support interface provides general design principles for engineering next-generation catalysts for sustainable fuel production.

Identifying and mechanistically understanding active sites is essential for elucidating reaction mechanisms and enabling rational catalyst design in heterogeneous catalysis[1,2]. Supported metal catalysts have been extensively used in various catalytic systems in the industrial process[3–5]. The supports are typically not inert as the carrier, and the interaction with the nanoparticles gives rise to new interfacial phenomena[6–8]. Such metal-support interaction (MSI) is supposed to have a profound impact on the resulting catalyst's structure formation,

[1]School of Chemistry and Chemical Engineering, Nanjing University of Science and Technology, Nanjing, China. [2]Department of Chemical Engineering, Norwegian University of Science and Technology, Trondheim, Norway. [3]Key Laboratory of Marine Chemistry Theory and Technology, Ministry of Education, College of Chemistry and Chemical Engineering, Ocean University of China, Qingdao, China. [4]Frontiers Science Center for Deep Ocean Multispheres and Earth System, and Key Laboratory of Marine Chemistry Theory and Technology, Ministry of Education, Ocean University of China, Qingdao, China. [5]College of Chemistry and Chemical Engineering, Xi'an Shiyou University, Xi'an, China. [6]School of Chemistry and Chemical Engineering, Hunan University of Science and Technology, Xiangtan, China. [7]Institute of Functional Nano & Soft Materials Laboratory, (FUNSOM), Jiangsu Key Laboratory for Carbon-Based Functional Materials & Devices, Joint International Research Laboratory of Carbon-Based Functional Materials and Devices, Soochow University, Suzhou, China. [8]Suzhou Institute of Nano-Tech and Nano-Bionics, Chinese Academy of Sciences, Suzhou, China. [9]These authors contributed equally: Hongfei Ma, Chen Chen, Guoyan Ma. ✉e-mail: hongfei.ma@njust.edu.cn; ycui2015@sinano.ac.cn; de.chen@ntnu.no

including the active center and the coordination environment[9–12]. Hence, reasonable strategies to manipulate the active center for optimized catalytic performance are highly desirable. In heterogeneous catalysis, the active centers and MSI can not only affect the structure and chemical state of the working catalysts, but also can influence the catalytic performance, including activity, selectivity, and stability[13]. And thus, regulating the interaction between the active center and the support is supposed to provide options for rational catalyst design, followed by higher catalytic efficiency.

Catalytic conversion of lignin to value-added chemicals is one typical complex process, in which multiple types of reactions are involved[14–19]. Additionally, it has significant meaning to upgrade lignin to green and clean fuel to meet the requirements of the circular economy. Hydrodeoxygenation (HDO) reaction is one of the most promising strategies to convert lignin and biomass to value-added products through the C-O bond cleavage reaction[20,21]. Lignin conversion can, in principle, yield a broad spectrum of valuable products, notably chemicals and biofuels[22–25]. Thus, it makes controlling the selectivity of the desired product more meaningful, despite the inherent challenge posed by the complex network of concurrent and sequential reactions involved in the upgrading process. Therefore, selectively converting lignin into specific hydrocarbons through tailoring catalyst design remains a key challenge and needed to be addressed. The most viable approach to achieve this goal needs the precise manipulation of the active center, given the complexity of the reaction systems and the competing reaction pathways involved in the HDO process[26–28]. Hence, more focus should be paid on the active site restructuring and regulation, and then convert lignin to the value-added products with higher yield.

In this work, we report the self-reaction-induced strong metal-support interaction on carbon nanofibers (CNF), driven by the surface restructuring between hydroxyl groups and leading to the formation of the interfacial Ru/RuOx heterostructures. The self-engineered manipulation contributes to the formation of well-defined bifunctional active sites comprising metallic Ru and partially oxidized RuOx at the interface. The resulting Ru/RuOx heterostructures exhibit good catalytic performance for the HDO of lignin-rich corncob, delivering a liquid hydrocarbon mass yield of 49.1% and a carbon yield of 67.7%. Notably, the interfacial Ru/RuOx heterostructure is capable of selectively transforming lignin into solely cycloalkanes. When oxygenated species are removed from the CNF surface, aromatic products can be produced. Additionally, aromatic compounds are predominantly produced when using the benchmark catalysts, Ru/Al2O3 and Ru/AC (AC: activated carbon) catalysts. Our results underscore the pivotal role of the interfacial Ru/RuOx site in tuning both the reaction pathways and the product selectivity. These insights offer significant implications for biomass upgrading, thereby promoting the generation of liquid hydrocarbons from sustainable feedstocks. Furthermore, this study elucidates the mechanisms through which catalysts can be fine-tuned to direct the reaction pathways and selectivity of the products.

## Results and discussion

Lignin is an abundant source of renewable aromatics that has long been utilized for valorization[29,30]. Currently, the traditional method employs the two-step strategy, in which lignin is firstly converted to oligomers by pyrolysis at relatively high temperature and pressure, and then to chemicals or fuels, as shown in Fig. 1a. However, the reactive oxygenated intermediates in bio-oils tend to undergo intermolecular condensation reactions and form chemically stable (poly) aromatic macromolecules that hinder the subsequent reactions and block the active sites. Another strategy employed in this work is the one-pot catalytic conversion process, which streamlines lignin valorization by integrating depolymerization and hydrodeoxygenation (HDO) into a single step. In this study, we focus on this one-pot approach to directly convert lignin-rich corncob into value-added liquid hydrocarbons.

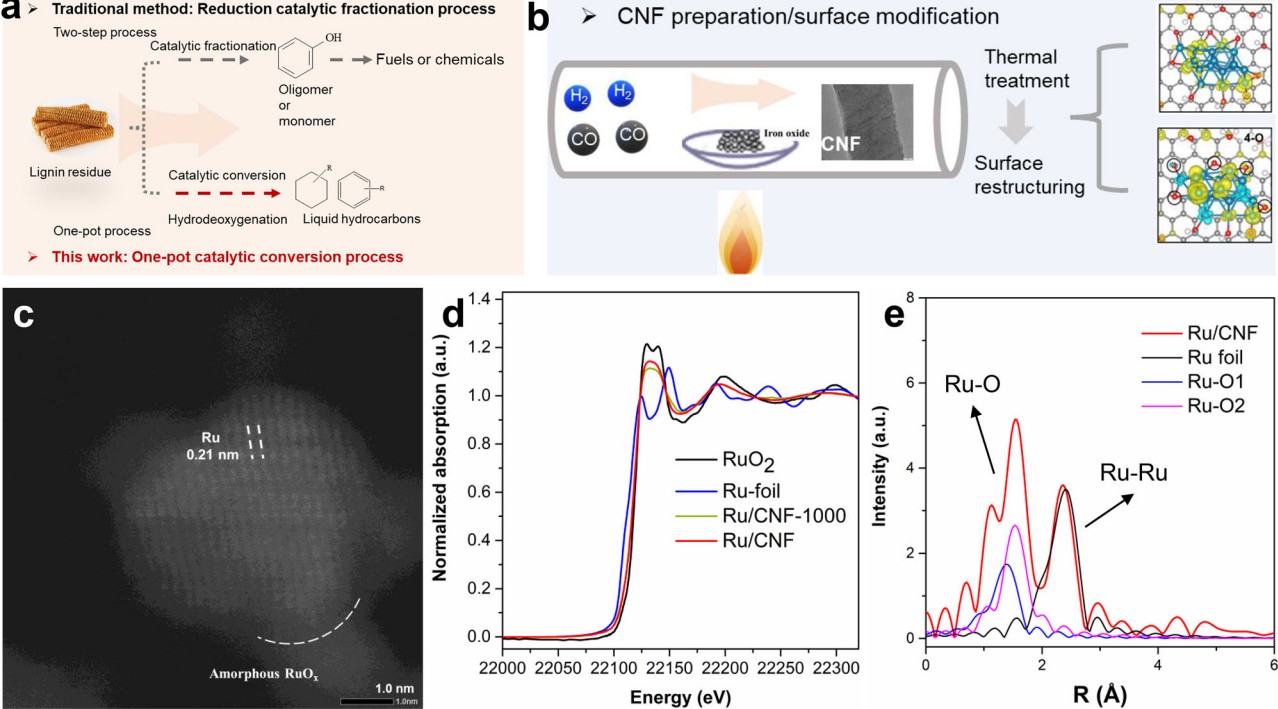

**Fig. 1 | Illustration of the methods used for the HDO reactions. a** Comparison of traditional method with this study for the lignin conversion, **b** Methods used in this work to synthesis carbon nanofiber, **c** HAADF-STEM images of Ru/CNF catalyst, **d** Ru K-edge XANES spectra of Ru/CNF catalysts and references, **e** FT-EXAFS spectra of Ru K-edge of Ru/CNF and Ru foil and RuO2.

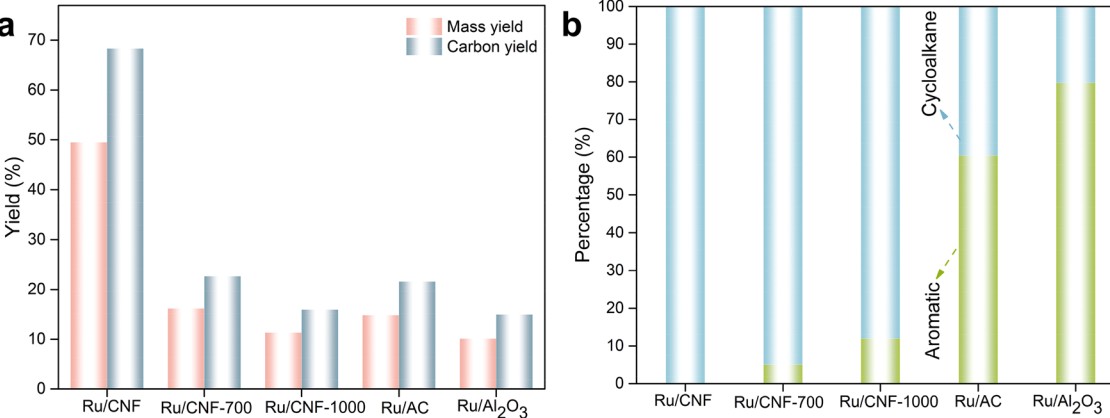

**Fig. 2 | Catalytic performance of HDO of lignin-rich corncob. a, b** Comparison of mass and carbon yields, product distribution of different catalysts at identical reaction conditions. Reaction conditions: 0.1 g lignin, $W_{cat}$ = 0.1 g, 20 ml dodecane, 5 MPa $H_2$ at 250 °C, 8 h.

Carbon nanofibers (CNFs), as catalyst supports, offer a range of advantages in catalysis due to their unique structural, chemical, and physical properties compared to conventional supports such as metal oxides and activated carbon (AC). These advantages include high surface area, excellent electrical conductivity, good chemical stability, and a flexible surface that can be functionalized to enhance the interaction with the active metal species. CNFs are prepared by the traditional chemical vapor deposition method[31,32], using iron oxide as the catalyst and syngas (CO/$H_2$: 4/1) as the carbon-containing gas, as shown in Fig. 1b. It has been commonly reported that oxygen species on the carbon surface usually play significant roles in determining the structure of the active sites, and then in the catalytic performance[33–35]. And thus, the as-synthesized CNF was thermally treated at varied temperatures (700 and 1000 °C) in argon to remove the oxygen species. Therefore, CNFs with varied oxygen species on the surface were obtained and used as the support for Ru catalysts. The physical properties of Ru/CNF catalyst pre-treated at different temperatures show similar pore structures (Table S1 in the Supplementary information). X-ray diffraction (XRD) patterns show that Ru is highly dispersed on the CNF support, since no Ru species can be observed, except for the carbon species (Fig. S1). The microstructure of the synthesized Ru/CNF was investigated by high-resolution transmission electron microscopy (HR-TEM). Energy-dispersive X-ray spectrometry (EDX) elemental mapping revealed the homogeneous distribution of Ru on the CNF. High-angle annular dark-field scanning transmission electron microscopy (HAADF-STEM) image in Figs. 1c and S2 reveals amorphous $RuO_x$ and crystalline Ru domains, with lattice fringes at 0.21 nm corresponding to the Ru (101) plane[36]. To investigate the electronic structure and coordination configuration of Ru, X-ray absorption near-edge structure (XANES) and extended X-ray absorption fine structure (EXAFS) spectroscopy measurements were performed at the Ru K-edge, as shown in Fig. 1d and e, and Table S2. The white-line intensity of Ru in Ru/CNF lies between that of $RuO_2$ and Ru foil, suggesting the presence of partially oxidized Ru species in the heterostructure catalyst. Figure 1e shows the phase-corrected Fourier transform (FT) curves at the R space of the Ru/CNF in comparison with the references of Ru foil and $RuO_2$. Notably, Ru/CNF has two dominant peaks corresponding to Ru-O and Ru-Ru path, confirming the state of mixed metallic Ru and partially oxidized $RuO_x$ state, consistent with the HAADF-STEM observation.

In addition, $Al_2O_3$ and activated carbon (AC) were also chosen as the benchmark catalysts in this work. The basic characterizations of physical adsorption, XRD, and high-resolution transmission electron microscopy (HR-TEM) of the Ru/AC, Ru/$Al_2O_3$, Ru/CNF, Ru/CNF-700, and Ru/CNF-1000 catalysts are shown in Table S1, and Figs. S3–S8 in the Supplementary information.

All catalysts exhibit a mesoporous structure, with Ru/AC showing the highest surface area and larger pore volume than Ru/CNF. Ru/$Al_2O_3$ also displays a higher pore volume than Ru/CNF. High-temperature treatment has a negligible effect on the Ru/CNF structure. XRD reveals no detectable Ru or $RuO_x$ diffraction peaks, only features from the carbon support, indicating high Ru dispersion across the carbon supports. TEM confirms the uniform Ru dispersion on CNF, AC, and $Al_2O_3$, demonstrating effective metal dispersion irrespective of support type.

During the enzymatic hydrolysis of corncobs for sugar production, a lignin-rich solid residue is generated as a primary byproduct, as illustrated in Fig. S9. This residue, commonly classified as industrial waste in the fine chemical industry, contains a significant proportion of lignin. Typically, lignin-rich residue can be depolymerized into lower molecular weight phenolic compounds, such as monomers or oligomers, which can subsequently be upgraded via HDO to produce liquid hydrocarbons or biofuels. In this study, the aforementioned corncob residue will be utilized and evaluated as a feedstock in the HDO process, as depicted in the following section.

The HDO reaction of the lignin-rich corncob residue was investigated over the various Ru catalysts, and the results are displayed in Fig. 2. Ru/CNF shows very high mass yield and carbon yield (49.1% and 67.7%) in direct conversion of lignin via HDO reaction. As summarized in Table S3, the hydrocarbon yield reported here represents one of the highest values achieved to date for converting lignin or woody biomass into liquid hydrocarbons. In addition to the yield, we would like to modestly highlight another aspect of the Ru/CNF catalyst for producing cycloalkanes. The one-pot selective transformation of lignin into cycloalkanes using hydrogen, as illustrated in Fig. S10, holds promise for a unified approach to hydrogen storage and the production of sustainable aromatics. In this context, cycloalkane can serve as an effective liquid organic hydrogen carrier[37,38], enabling the transportation and on-site release of hydrogen via the integration with the dehydrogenation process. This process underscores the capability of the Ru/CNF catalyst to convert lignin into environmentally friendly and commercially valuable products.

Ruthenium-supported catalysts have demonstrated high activity in the HDO of the lignin model compounds[39]. For comparison, we selected Ru/$Al_2O_3$ and Ru/AC catalysts as benchmark catalysts. The Ru/AC catalyst exhibited slightly higher yields than Ru/$Al_2O_3$, with the mass and carbon yields of 14.8% and 21.6%, respectively. Importantly, the choice of these benchmarks also reflects differences in surface hydroxyl mobility: while hydroxyl groups on $Al_2O_3$ are strongly bound and immobile, limiting surface restructuring, the mobile hydroxyl species on CNFs enable the formation of the interfacial Ru/$RuO_x$ heterostructures under thermal treatment, which we identify as key to the enhanced HDO performance. In contrast, the Ru/CNF catalyst

developed in this work achieved significantly higher yields than the two benchmark catalysts. Beyond the yield difference, the product distributions also varied notably depending on the catalyst supports (Fig. 2b). Notably, no aromatic products were detected on Ru/CNF, indicating that all the unsaturated C = C bonds, including the aromatic $C_6$ ring, were hydrogenated to C-C bonds, with only cycloalkane hydrocarbons formed. In contrast, the main products of the benchmark catalyst were aromatics. The Ru/AC catalyst showed a lower aromatic distribution, suggesting a moderate hydrogenation capability compared to Ru/Al$_2$O$_3$. The detailed product distribution and yields with regard to the carbon numbers are listed in Fig. S11 and Table S4. It can be seen that only cycloalkanes are produced, with $C_6$ as the dominating one on the Ru/CNF catalyst. However, it is reversed on the benchmark catalysts, Ru/AC and Ru/Al$_2$O$_3$, with $C_6$-$C_8$ aromatics as the dominating products. The results demonstrate the significant roles in influencing the C-C bond cleavage in the HDO process. This integrated depolymerization and upgrading approach provides an effective route for one-pot HDO of lignin to liquid cycloalkanes. (illustrated in Fig. S12).

Catalyst support is playing a crucial role in governing both HDO activity and product distributions. Notably, CNF demonstrates superior performance to conventional activated carbon, highlighting the importance of support characteristics. The electronic properties of carbon materials are strongly dependent on their surface functional groups. To probe the structure–activity relationship, the CNF supports were thermally treated at 700 °C and 1000 °C (Ru/CNF-700 and Ru/CNF-1000), progressively eliminating surface oxygen functionalities.

This thermal treatment of the CNF led to significant changes in catalytic performance. Compared to untreated Ru/CNF, the high-temperature-treated catalyst showed markedly reduced mass yields (16.2% for Ru/CNF-700 and 11.5% for Ru/CNF-1000) and carbon yields. More interestingly, we observed a temperature-dependent emergence of aromatic products, with yields increasing progressively to reach 12% for Ru/CNF-1000. These results clearly demonstrate that high-temperature treatment modifies the chemical environment of the active site by removing oxygen functional species, thereby enabling precise tuning of product distributions. The detailed mechanistic implications of these findings will be discussed in the following section.

To assess the catalyst stability, we performed recycling tests under semi-continuous conditions by replenishing fresh lignin feedstock without separating the catalyst, solvent, and products between cycles at the same optimized reaction conditions. The Ru/CNF catalyst maintained consistent activity over six consecutive runs without observable deactivation (Fig. S13) for both batch and continuous operation modes.

To evaluate the practical applicability, we extended our investigation to real beechwood biomass (sourced from Dansk Traemel) as a feedstock for biofuel production via HDO reaction. The catalyst exhibited robust performance, achieving a hydrocarbon mass yield of approximately 20% with complete deoxygenation (Fig. S14). Although the yield is lower than that obtained with lignin-rich corncob, this difference likely arises from the significantly higher oxygen content in the raw biomass. Importantly, the product distribution aligned with previous findings, consisting exclusively of hydrocarbons, including both aromatics and cycloalkanes, with no detectable oxygenated species. To further assess the scalability of the one-pot conversion process from lignin to liquid hydrocarbons, a scale-up experiment was performed by increasing the feedstock amount fivefold, as shown in Fig. S15. A mass yield of 44% was achieved, with cycloalkanes as the main products and no detectable oxygenates, underscoring the scalability of the process. Together, these results conclusively demonstrate that the Ru/CNF catalyst maintains high HDO activity across both lignin-rich corncob and complex, real-world woody biomass. Its consistent performance, full deoxygenation capability, and

adaptability to scaled reaction conditions underscore its potential as a robust and practical catalyst for sustainable biofuel production.

The spent catalysts after HDO reaction were also characterized to verify the stability in this complex reaction condition. XRD profiles (Fig. S16a) revealed no detectable Ru-species diffraction peaks, while X-ray photoelectron spectroscopy (XPS) analysis of the Ru 3p region (Fig. S16b) showed nearly identical binding energies and species distributions between the fresh and spent catalysts, indicating excellent preservation of the Ru chemical state during the HDO reaction process. HR-TEM confirmed the absence of metal particle agglomeration (Fig. S17). The comprehensive characterization results confirm that the Ru/CNF catalyst exhibits outstanding structural and chemical stability while maintaining high catalytic performance for lignin conversion to cycloalkanes in semi-continuous HDO processes.

To understand the reaction mechanisms over the various catalysts, kinetic studies were conducted. As shown in Figs. S18 and S19, increasing temperature enhances the mass yield over Ru/CNF, in line with the typical temperature dependence of HDO reactions. At lower temperatures, a substantial presence of oxygenated intermediates with saturated aromatic ring-like cyclohexanol and methoxy cyclohexane was produced on the Ru/CNF catalyst with a relatively low yield. The oxygenated species, constituting 30.8% of the products, with the saturated benzene ring, was partially deoxygenated. Time-dependent experiments conducted at 250 °C on the Ru/CNF catalyst, illustrated in Figs. S20–S22, revealed the formation of oxygenated products at the earlier stage of the reaction. These intermediates were gradually transformed into hydrocarbons over time, with no aromatic compounds detected under these conditions. However, the time-dependent experiment on the Ru/AC catalyst revealed a distinct behavior (Fig. S23), where aromatic products dominate even in the early stages of the reaction. These results suggest that the Ru/CNF and Ru/AC catalysts follow different reaction pathways, particularly in the initial phase, during the HDO process.

To gain a comprehensive understanding of the local structure of Ru and its interaction with oxygen species on the CNF support, near-ambient pressure XPS (NAP-XPS) was performed. The samples were reduced under H$_2$ to mimic the reducing atmosphere of the HDO process, while ensuring they were not exposed to air during transfer or measurement. Nevertheless, we recognize that 0.2 mbar is nearly four orders of magnitude below the 5 MPa H$_2$ pressure used in catalytic testing, and the catalyst surface under high-pressure H$_2$ is likely to be more reduced than our NAP-XPS measurement indicates. However, the systematic variation in oxidation states observed at different pre-treatment temperatures under NAP-XPS conditions is expected to reflect the relative trend that persists under reaction conditions, even if the absolute values differ. The C1s and Ru 3 d spectra of three samples, namely Ru/CNF, Ru/CNF-700, and Ru/CNF-1000, after reduction in H$_2$ for 1 h during the heating process, are illustrated in Fig. 3. The C 1 s spectra are shown in Fig. 3a; the two peaks can be assigned to C-C and C-O bonds[33]. When Ru is supported on the CNF, the C 1 s spectra overlap with Ru 3 d spectra, as shown in Fig. 3b, illustrating four main peaks, in which two more peaks correlated to Ru species exist. Besides, when high-temperature treatment is carried out on the CNF support to remove the oxygen species, the binding energy of Ru shifted to lower values[40]. It indicates a strong metal-support interaction between Ru and CNF on Ru/CNF.

The Ru 3 d spectra were further deconvoluted, as shown in the inset in Fig. 3b. It can be seen that two peaks can be observed on all the Ru/CNF catalysts, which can be ascribed to metallic Ru ( ~ 280.1 eV) and oxidized Ru ( ~ 280.6 eV). It confirms the existence of a heterostructure catalyst consisting of metallic Ru and ruthenium oxide (RuO$_x$). Moreover, the contributions of Ru and RuO$_x$ are also calculated. The contributions of RuO$_x$ are 58%, 49.9%, and 35.2% on Ru/CNF, Ru/CNF-700, and Ru/CNF-1000. It is consistent with the high-resolution Ru 3p spectra (Fig. S24 and Table S5) that the peak deconvolution

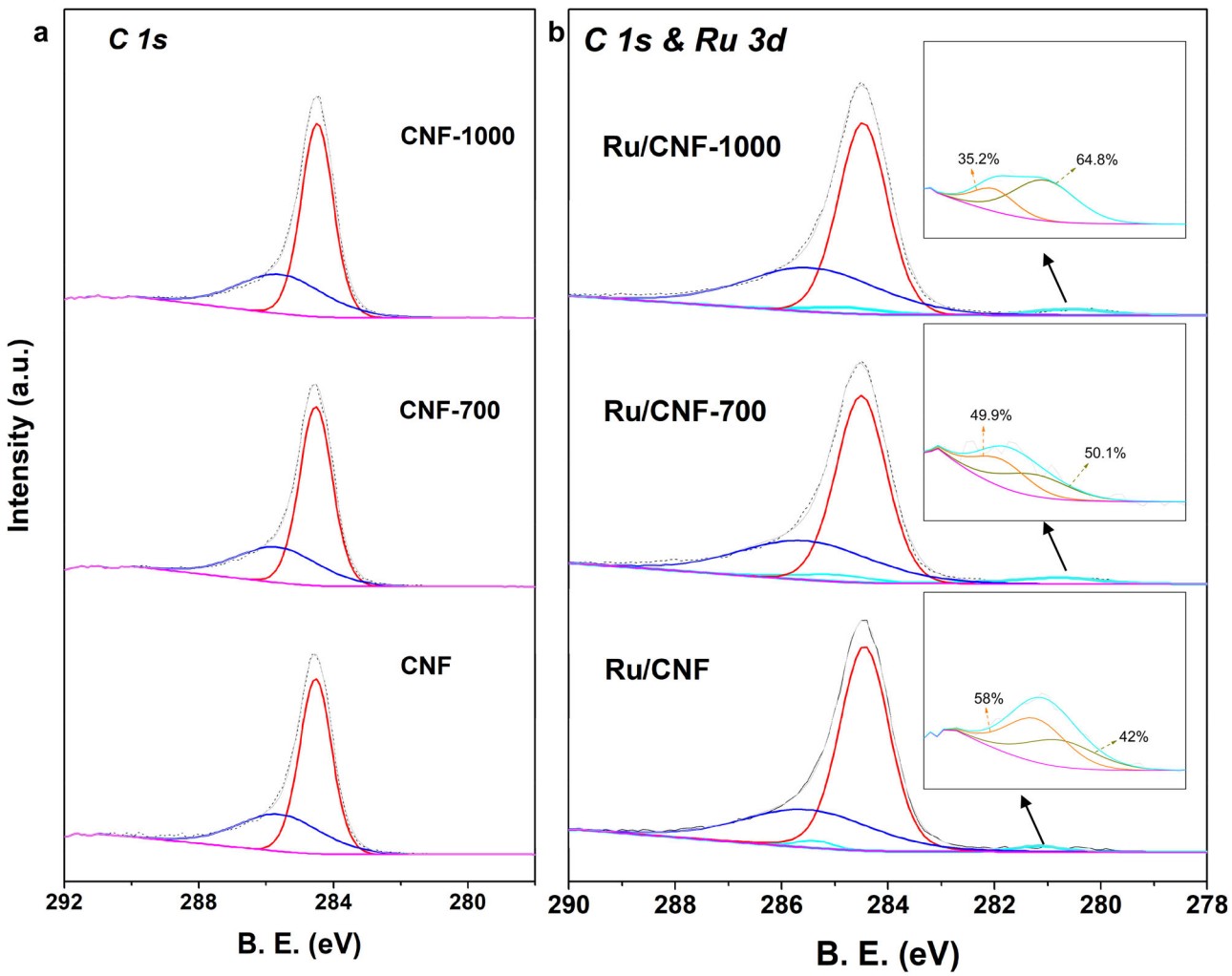

**Fig. 3 | Near ambient pressure X-ray photoelectron spectra (NAP-XPS). a** CNF and **b** Ru/CNF catalysts at 300 °C under the $H_2$ atmosphere (0.2 mbar) for 1 h.

discoveries. The contribution of $RuO_x$ on Ru/CNF is the highest. It demonstrates that the oxidized form is decreased by removing the oxygen functional species. The results showcase that the oxygen functional groups on CNF are playing significant roles in affecting the interaction between the active site and CNF support. In the following part, we will discuss the intrinsic mechanism for the oxygen-decorated Ru site (or partially oxidized Ru) evolution and reaction pathway of the HDO process.

The formation of $Ru/RuO_x$ interface is also confirmed by the high-resolution of O 1 s XPS (Fig. S25 and Table S6). For Ru/CNF, the high-resolution O 1 s spectrum deconvolutes into four peaks. The peak at 530.8 eV (Peak 1) corresponds to the partially oxidized Ru species, which is absent in the CNF support. The remaining peaks are attributed to oxygen-containing surface groups (hydroxyl, ester, and carboxyl). The existence of Ru-O species is possibly induced by the surface reaction between mobile oxygen functional groups.

To elucidate the nature of active sites and support effects in the HDO reaction, first-principles calculations were also carried out. The simulations tracked the temperature-dependent evolution of surface hydroxyl groups, aligning with experimental observations that OH species are progressively removed as temperature increases. To present catalysts subjected to different thermal treatments, we constructed models with systematically varied OH coverages on the carbon support. Figures 4a and b depict the initial states and the optimized structures with different OH group densities, respectively. The OH-rich model corresponds to the oxidized Ru/CNF catalyst,

whereas the OH-deficient model reflects the thermally treated Ru/CNF-700 surface.

Under thermal treatment, the OH-enriched carbon surface undergoes structural restructuring. The hydroxyl groups on carbon exhibit high mobility and tend to migrate toward the Ru nanoparticles, driven by the strong Ru-O affinity. Upon adsorption onto the Ru surface, adjacent OH species recombined together to form O* and $H_2O^*$ via the following equation:

$$2OH^* \rightarrow O^* + H_2O^* \qquad (1)$$

The desorption of $H_2O^*$ leaves behind strongly bound O* species at the Ru-support interface, triggering significant local structural reconstruction. This process partially oxidizes metallic Ru, generating partially oxidized $RuO_x$ sites and yielding a bifunctional surface comprising coexisting metallic $Ru^0$ and oxidized $Ru^{++}$ domains, consistent with the XPS observations.

Molecular dynamics (MD) simulations further corroborate this transformation. The energy profiles for Ru/CNF and Ru/CNF-700 (Figs. 4c and d, and S26) reveal that neighboring OH groups on Ru can be coupled to form O* and $H_2O^*$, followed by $H_2O$ desorption, leaving behind a stable partially oxidized $RuO_x$ site. The inset depicts the transition state (TS) structures, where two OH groups combine into $H_2O^*$ and O*, with a blue circle highlighting the desorbing $H_2O$ molecules. The energy was found to oscillate around a constant frequency due to the surface reactions. The Ru-O bond length was also

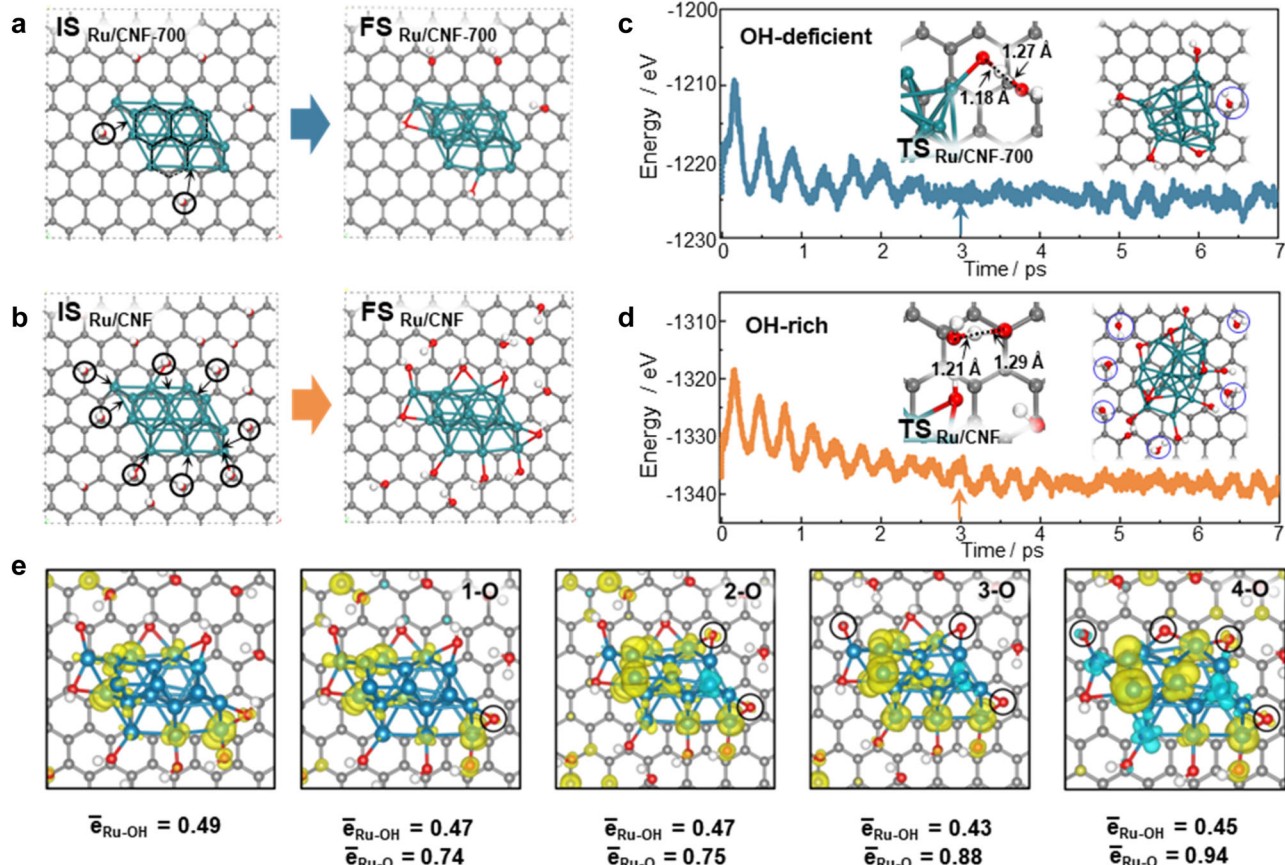

**Fig. 4 | MD simulation of the Ru/CNF catalysts. a** Initial state (IS) and final state (FS) structures of the Ru/CNF-700 model before and after optimization. Black hexagons represent the lattice alignment between Ru clusters and the carbon nanolayer. **b** IS and FS structures of the Ru/CNF model before and after optimization. **c** Energy profile of the molecular dynamic (MD) simulation for Ru/CNF-700 from 0 ps to 7 ps at 573 K. **d** Energy profile of the MD simulation for Ru/CNF over the same time range and temperature. The inset shows the transition state (TS) structure of two OH groups coupling to form one $H_2O$ and one O atom, with the blue circle highlighting the $H_2O$ molecule desorbing from the surface. The arrow indicates the approximate point where the energy and temperature begin to oscillate around a constant value. **e** Average Bader charge of Ru atoms bonded to OH and O species at varying surface coverages on the Ru surface. Green: Ru; Grey: C; Red: O; White: H.

calculated, as shown in Fig. S26. The calculated Ru-O bond lengths show that Ru-O in Ru/CNF (1.706 Å) is shorter than that in Ru/CNF-700 (1.806 Å), further corroborating the strong metal-support interaction observed in the XPS.

Bader charge analysis was carried out to elucidate how surface OH and O species, located exclusively near Ru and on the carbon surface, modify the electronic structure of the Ru (Fig. 4e). Both OH and O withdraw electron density from adjacent Ru atoms, OH groups induce only modest changes, leading to a slight increase in the local positive charge of the directly bonded Ru sites. In contrast, O atoms exert a much stronger electron-withdrawing effect: interfacial Ru atoms bonded to O become markedly electron-deficient, and this depletion extends across the Ru cluster surface, generating a pronounced gradient in oxidation states. As O* coverage increases, the entire Ru cluster becomes progressively polarized, while the surface O atoms accumulate electron density. This cooperative charge redistribution gives rise to highly polarized $Ru^{\delta+}\cdots O^{\delta-}$ pairs, giving rise to electronically polarized Ru−O interfacial ensembles capable of heterolytic-like $H_2$ activation. In this configuration, electron-deficient Ru atoms function as Lewis acids, while electron-rich O atoms serve as Lewis bases. Concurrently, the Ru cluster exhibits an oxidation-state gradient, resulting in a heterogeneous surface charge distribution. All together it creates a bifunctional interface capable of enhancing reactant activation and accelerating key surface reactions.

Overall, the MD calculation qualitatively captures the experimental observation of Ru oxidation-state changes with the oxygen content of carbon through tuning the pretreatment temperature. It provides deep insights into the long-observed fact in the literature that the oxygen groups on the carbon can significantly influence the charge of the metal clusters[34,35,41]. Calculations reveal that on OH-enriched surfaces, the surface OH groups migrate from the carbon support to the Ru clusters, generating partially oxidized Ru sites at the Ru-carbon interface as the dominant configuration on initially OH-rich Ru/CNF. This restructuring increases the positive Bader charge of the surface Ru atoms, indicating the development of electron-deficient Ru centers. These trends are consistent with the Ru 3 d XPS spectra, in which Ru/CNF exhibits the highest binding energy among all CNF-supported catalysts, confirming the electron-withdrawing effect of oxygen-decorated Ru sites. The correlation between higher Ru 3 d binding energies and the calculated positive Bader charges underscores the critical role of surface O/OH species in modulating Ru electronic properties and strengthening the interaction with oxygenates during the HDO reaction.

The HDO reaction mechanism involves three critical elementary steps: hydrogen activation and dissociation, C-O bond cleavage, and aromatic ring hydrogenation. DFT calculations were carried out to elucidate how surface structure modulates these reaction steps by comparing three representative Ru/CNF structures-$Ru_{18}(OH)_5$ (OH-deficient), $Ru_{18}(OH)_{16}$ (OH-rich), and $Ru_{18}(OH)_8O_4$ (O-rich) models.

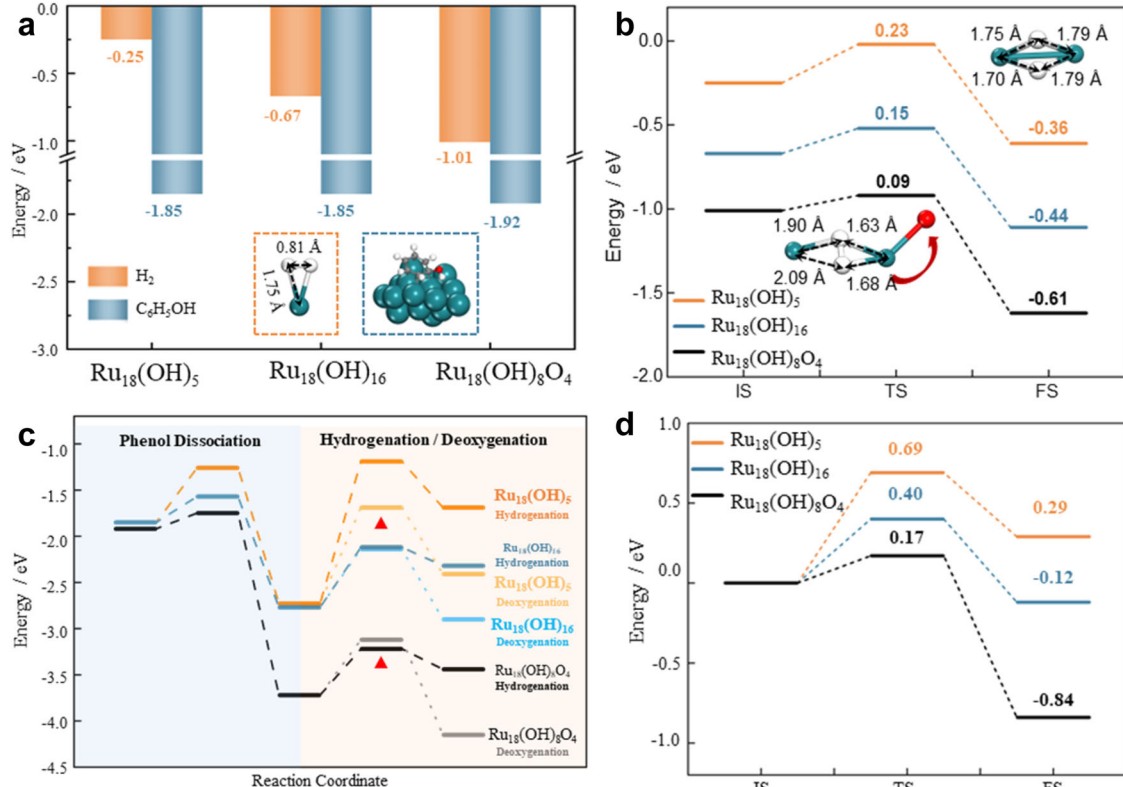

**Fig. 5 | Gibbs free energy calculations of the crucial steps in HDO. a** Adsorption energies of $H_2$ and phenol on $Ru_{18}(OH)_5$ (OH-deficient), $Ru_{18}(OH)_{16}$ (OH-rich) and $Ru_{18}(OH)_8O_4$ (O-decorated) models. The insets display the adsorption configurations of $H_2$ and phenol, with bond lengths indicated in black. **b** Energy profiles for $H_2$ dissociation reactions, with insets showing the transition state structures. The H–H bond length at the transition state is indicated in black. **c** Energy profiles for

phenol dissociation and subsequent hydrogenation or deoxygenation processes. The red triangles denote the preferred reaction pathway. The inset illustrates the transition state configuration for the deoxygenation reaction. **d** Energy profiles for the deoxygenation reaction of cyclohexanol. The inset depicts the transition state configuration. Green: Ru; Grey: C; Red: O; White: H.

These models vary in surface OH/O coverage and Ru−O coordination and are constructed to represent the key structural features of the catalysts. Instead, they serve as quantitative descriptors of the surface evolution induced by thermal pretreatment, capturing the transformations from OH-rich surface at low pretreatment temperature to O-rich surface at moderate temperatures and ultimately to OH-deficient surface after high-temperature treatment. The adsorption energies and the energy profiles for hydrogen activation, phenol dissociation, hydrogenation, and phenol deoxygenation over the three Ru/CNF models-$Ru_{18}(OH)_5$, $Ru_{18}(OH)_{16}$, and $Ru_{18}(OH)_8O_4$-presented in Fig. 5, while the corresponding transition-state and intermediate structures are summarized in Table S7.

Hydrogen adsorption and activation on the three Ru/CNF model surfaces are strongly influenced by the surface oxidation state and the extent of Ru-O coordination (Fig. 5a, transition state structures in Table S7). Calculated $H_2$ adsorption energies increase with oxygen coverage: −0.25 eV on OH-deficient $Ru_{18}(OH)_5$, −0.67 eV on OH-rich $Ru_{18}(OH)_{16}$, and −1.01 eV on O-decorated $Ru_{18}(OH)_8O_4$. This trend arises from progressive electron withdrawal from surface Ru atoms toward adjacent O species, generating Ru sites with varying degrees of electron deficiency (Table S8). The resulting $Ru^{\delta++}-Ru^{\delta+}$ pairs, while both Lewis acidic, exhibit sufficient charge asymmetry to polarize the H-H σ bond, enabling cooperative $H_2$ activation analogous to a Lewis acid-Lewis acid pair.

Consistent with this interpretation, the activation barrier for $H_2$ dissociation follows the inverse trend: $Ru_{18}(OH)_5$ (0.23 eV) > $Ru_{18}(OH)_{16}$ (0.15 eV) > $Ru_{18}(OH)_8O_4$ (0.09 eV), with the O-rich surface also yielding the most stable dissociated state (−0.61 eV). The barrier appears to be governed by the degree of charge asymmetry within

adjacent Ru pairs. On a metallic Ru surface, $H_2$ activation proceeds homolytically via σ-donation/back donation, limited by d-band filling and resulting in moderately high barriers. The transition state on such surfaces shows a nearly symmetric H-H bond length (~1.7–1.9Å). In contrast, on the O-decorated surface, electron withdrawal by oxygen generates a built-in electric field that promotes heterolytic polarization. This is reflected in the asymmetric transition-state geometry, where the Ru-H bond adjacent to oxygen is significantly shorter (~1.63–1.68 Å), and the opposite Ru-H bond is elongated (~2 Å). These observations suggest that the Ru-O interface plays a decisive role in driving $H_2$ activation at the $Ru/RuO_x$ boundary.

Phenol adsorbs on Ru surfaces primarily through its aromatic π system (Fig. 5a, Table S7). Adsorption energies are similar on $Ru_{18}(OH)_5$ (−1.85 eV) and $Ru_{18}(OH)_{16}$ (−1.85 eV), with slightly stronger binding on $Ru_{18}(OH)_8O_4$ (−1.92 eV). This correlates with increasing Ru electron deficiency, which in turn coincides with lower calculated barriers for O-H dissociation. In this step, O-H activation proceeds on polarized Lewis acidic Ru sites via a transition state of the form $C_6H_5O\cdots H\cdots Ru^{\delta+}-O^{\delta-}$ ($OH^\delta$+), a detailed analysis is provided in Supplementary Note 1 of the supporting information.

The barrier for phenolic O-H dissociation reflects the degree of surface polarization imparted by oxygen functionalities (Fig. 5c). The O-rich surface exhibits the lowest barrier (0.17 eV), where strongly electron-withdrawing O ligands generate a pronounced $Ru^{\delta+}-O^{\delta-}$ charge asymmetry that efficiently polarizes and cleaves the O-H bond. The OH-rich surface shows a moderate barrier (0.28 eV), likely because pre-adsorbed OH groups partially neutralize Ru-O polarization and occupy interfacial sites. In contrast, the OH-deficient surface, lacking interfacial oxygen, remains largely metallic and provides insufficient

charge separation, resulting in the highest barrier (0.59 eV). This pattern suggests that O-induced surface polarization is important for efficient phenol activation.

Following phenoxy formation, two competing HDO pathways are possible[42,43]: (i) direct deoxygenation of the phenoxy intermediate to aromatic hydrocarbons and (ii) ring hydrogenation to cyclohexanol followed by deoxygenation to cycloalkane. The energy barriers of the two pathways are also compared, as shown in Fig. 5c. The difference in the energy barrier between ring hydrogenation and direct deoxygenation reflects the selective production of cyclohexane and aromatics.

The benzene-ring hydrogenation step in phenol HDO shows a strong dependence on the surface polarity generated by oxygen functionalities on Ru/CNF. DFT results reveal that the O-rich $Ru_{18}(OH)_8O_4$ surface exhibits the lowest hydrogenation barrier (0.50 eV), followed by the OH-rich $Ru_{18}(OH)_{16}$ surface (0.65 eV), while the OH-deficient $Ru_{18}(OH)_5$ surface shows a much higher barrier (1.54 eV), shown in Fig. 5c. The transition state structures are illustrated in Table S7. This trend mirrors the degree of charge asymmetry created at the $RuO_x$ interface, see detailed discussion in Supplementary Notes 2 and 3. The higher Ru charge promotes the hydrogenation ability of H into the aromatic ring. As for the other reaction pathway, direct deoxygenation of phenoxy, the $Ru_{18}(OH)_8O_4$ surface also shows the lowest energy barrier (0.6 eV) compared with 0.63 eV for the $Ru_{18}(OH)_{16}$ and 1.04 eV for the $Ru_{18}(OH)_5$ catalysts, as shown in Table S6. It indicates that the direct deoxygenation reaction on $RuO_x$/Ru interfaces is also favorable in comparison with other catalysts.

Besides, we can compare the energy barrier of hydrogenation and direct deoxygenation reactions on each catalyst to derive the most possible reaction pathway of HDO process. On the $Ru_{18}(OH)_8O_4$ structure, the energy barrier of hydrogenation is much lower than deoxygenation (0.5 and 0.6 eV). The energy barrier of the deoxygenation reaction is slightly lower on the $Ru_{18}(OH)_{16}$ surface-0.65 eV for hydrogenation and 0.63 eV for deoxygenation. However, it is reverse for the $Ru_{18}(OH)_5$ surface, and it is 1.54 eV for the hydrogenation reaction and 1.04 eV for the deoxygenation reaction. The results indicate that the catalysts are following different kinetic behaviors, in which hydrogenation is more favorable for the O-decorated Ru/$RuO_x$ interface, and direct deoxygenation is more favorable for the $Ru_{18}(OH)_5$ surface. Therefore, the heterostructure of Ru/$RuO_x$ shows high activity for both hydrogenation and direct deoxygenation of phenoxy than other catalysts.

The energy barrier of deoxygenation of cyclohexanol is displayed in Fig. 5d and Table S7. The O-rich $Ru_{18}(OH)_8O_4$ surface displays the lowest energy barrier (0.17 eV), compared with 0.4 eV for the $Ru_{18}(OH)_{16}$ and 0.69 eV for the $Ru_{18}(OH)_5$ catalysts. It indicates that the deoxygenation of cyclohexanol is also more favorable on the O-decorated interface.

It has been commonly reported that hydrogenation of benzene is rather difficult, and high pressure is usually needed. The energy barrier of hydrogenation of benzene is illustrated in Fig. S27 and Table S7. The O-decorated Ru/CNF surface again shows the lowest barrier (0.75 eV) than $Ru_{18}(OH)_{16}$ (0.99 eV) and $Ru_{18}(OH)_5$ (2.12 eV) catalysts. It can be seen that hydrogenation of benzene on all the catalysts shows a higher energy barrier than the reactions we mentioned above. The O-decorated Ru/$RuO_x$ interface facilitates the dissociation of $H_2$, providing abundant H* species that are highly beneficial for the benzene hydrogenation reaction.

This work establishes a unified mechanistic and structural framework for how surface oxygen functionalities regulate HDO of lignin-rich corncob over Ru/CNF. By integrating experimental with DFT calculations, we show that thermally driven restructuring of surface hydroxyl groups on CNF produces partially oxidized, O-decorated interfacial heterostructure Ru/$RuO_x$. The adjacent oxygen ligands generate a heterostructure, characterized by $O^{\delta-}-Ru^{\delta++}-Ru^{\delta+}$ charge distribution. This interfacial polarization enhances heterolytic $H_2$ activation and promotes charge-separated interactions with phenolic −OH and C−O bonds. The interfacial heterostructure Ru/$RuO_x$ strongly favors the hydrogenation-deoxygenation route, enabling the conversion of lignin into fully saturated cycloalkanes with a mass yield of 49.1% and a carbon yield of 67.7%, whereas removal of surface oxygen shifts selectivity toward aromatic products. The structure-function relationships demonstrate that tuning oxygen coverage and the density of polarized Ru-O ensembles offers a rational strategy for controlling HDO selectivity. Beyond the Ru/CNF system, this concept provides a general design principle for creating multifunctional catalysts via deliberate modulation of metal-support interactions for biomass valorization and sustainable fuel production.

## Data availability
The authors declare that all data supporting the findings of this study can be found in the manuscript and Supplementary Information, or are available from the corresponding authors upon request.

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

## Acknowledgements

This work was financially supported by the European Union's Horizon 2020 research and innovation program (Grant Agreement No 101006744), the Fundamental Research Funds for the Central Universities under the contract number of No.30925010403. G. Ma acknowledges the financial support of the National Natural Science Foundation of China (No. 52204045) and Scientific Research Program Funded by Shaanxi Provincial Education Department (24JC070). The authors thank Prof. Y. Li (Aalto University) for providing the lignin. The authors also wish to thank the staff from Scientific Compass (www.shiyanjia.com) for providing invaluable assistance.

## Author contributions

H.M. designed and performed the catalyst preparation, catalytic reactions, and most of the characterizations, analyzed the data, and wrote the manuscript. C.C. performed the DFT calculation. G.M., T.O., Q.W., and W.W. provided helpful assistance on some of the characterizations and discussions. H. Z. and H.M. analyzed the XAS data. Y.L. and Y.C. performed the NAP-XPS tests. H.M. and D.C. conceived the idea, wrote, and revised the paper with feedback from the other authors. All the authors have given approval to the final version of the manuscript.

## Funding

 Olavs Hospital - Trondheim University Hospital).

## Competing interests

The authors declare no competing interests.
