## [Transparent Peer Review file · Nature Communications]

Interfacial Ru/RuO_x Heterostructures on Carbon Support Regulate Selectivity in Lignin Hydrodeoxygenation

Corresponding Author: Professor De Chen

Version 0:

Reviewer comments:

Reviewer #1

(Remarks to the Author)

The manuscript "Dynamic Ru/RuO_x Interface on Carbon Supports Regulate Selectivity in Lignin Hydrodeoxygenation" reports a combined experimental and computational investigation of Ru catalysts supported on carbon nanofibers for lignin hydrodeoxygenation (HDO), claiming that a dynamically restructured Ru/RuO_x interface governs product selectivity toward cycloalkanes versus aromatics. The work combines catalytic testing on lignin and real biomass feedstocks, near-ambient pressure XPS, and DFT/MD calculations to rationalize the observed trends. While the topic is timely and the experimental-theoretical coupling is ambitious, several key claims are currently overstated, particularly regarding the dynamic nature of the active interface and the mechanistic interpretation derived from the computational modeling. With a clearer definition of the active site, a more restrained mechanistic interpretation, and a rigorous comparison to the literature, the manuscript could become suitable for Nature Communications. I therefore recommend major revision. My main concerns are as follows:

1. A central claim of the manuscript is that a dynamic Ru/RuO_x interface regulates selectivity in lignin HDO. However, the experimental evidence primarily supports thermally induced, pretreatment-dependent interfacial states, rather than genuinely dynamic restructuring under reaction conditions. The near-ambient pressure XPS measurements are conducted under static H₂ at 300 °C and do not demonstrate reversible or reaction-driven evolution of the Ru oxidation state during catalysis. As presented, the data convincingly show controllable interface formation via CNF surface oxygen chemistry, but not dynamic interfacial behavior in the strict sense. Given the prominence of this concept in the title and abstract, the terminology should be moderated or further experimental evidence should be provided.

2. The manuscript claims outstanding performance and, in some instances, suggests record-level yields for lignin-to-hydrocarbon conversion. Such claims are difficult to substantiate without a rigorous, quantitative comparison to the literature. I strongly recommend the inclusion of a dedicated comparison table summarizing representative HDO catalysts reported in the literature, including lignin type, reaction conditions (temperature, pressure, solvent), catalyst composition, mass yield, carbon yield, and product selectivity (aromatics vs cycloalkanes). This would greatly strengthen the manuscript by clearly positioning the Ru/CNF system relative to existing approaches and avoiding overstatement.

3. The manuscript repeatedly attributes catalytic selectivity to surface frustrated Lewis pair (sFLP)-like ensembles involving O^{δ-}-Ru^{δ+}-Ru^{δ+} motifs. While the concept is appealing, the physical identity of the active site remains insufficiently resolved. It is not clear whether the dominant functionality arises from Ru-O perimeter sites, polarized Ru-Ru ensembles, or partially oxidized Ru edge atoms. Moreover, the distinction between the proposed sFLP-like behavior and more classical metal-oxide interfacial effects is not rigorously established. At present, the sFLP terminology appears more descriptive than demonstrative, and its use should be more carefully justified or framed more cautiously.

4. The DFT and molecular dynamics calculations are technically sound and internally consistent, and they provide a coherent mechanistic narrative linking surface oxygen chemistry to hydrogen activation and C-O bond cleavage. The MD simulations capturing hydroxyl migration and partial Ru oxidation at the Ru-carbon interface are particularly convincing and qualitatively consistent with the XPS observations. The construction of OH-rich, O-rich, and OH-deficient Ru₁₈ cluster models is a reasonable approach to rationalize pretreatment effects. However, the manuscript does not sufficiently emphasize the simplified and descriptor-based nature of the computational models. The chemistry is reduced to phenol as a lignin proxy, solvent effects and high-pressure hydrogen are neglected, only a single Ru cluster size is considered, and

coverage or competitive adsorption effects are not explored. These approximations are acceptable for extracting qualitative trends, but the manuscript occasionally implies a degree of quantitative correspondence with real lignin HDO that is not fully justified. The DFT results should be more explicitly framed as providing mechanistic trends and design principles rather than predictive descriptions of the full catalytic system.

5. The manuscript attributes selectivity primarily to differences in intrinsic reaction barriers induced by surface polarization at the Ru/RuO_x interface. While the DFT results are consistent with this interpretation, the experimental data do not fully disentangle intrinsic kinetic effects from potential influences of hydrogen availability, mass transport, or solvent-mediated stabilization. No site-normalized rates, kinetic isotope experiments, or competitive adsorption measurements are provided. As a result, the causal link between the calculated energy barriers and macroscopic selectivity remains suggestive rather than definitive and should be discussed with appropriate caution.

6. The manuscript is generally well written, but several sections—particularly those describing phenol activation and hydrogenation pathways—are repetitive between the main text and the Supplementary Information. In addition, the discussion of cycloalkanes as liquid organic hydrogen carriers, while interesting, feels peripheral to the core mechanistic message and could be shortened to improve focus.

Reviewer #2

(Remarks to the Author)

Overall Assessment:

This manuscript presents a Ru/CNF catalytic system for the one-pot hydrodeoxygenation (HDO) of lignin into cycloalkanes, proposing a "dynamic Ru–RuO interface" to explain performance variations. While the topic—integrating renewable resource conversion, interface catalysis, and metal oxidation state modulation—is timely and relevant to sustainable chemistry and biomass upgrading, the manuscript suffers from significant shortcomings in novelty, mechanistic depth, evidential rigor, and data quality. Despite a seemingly complete framework covering catalyst synthesis, characterization, performance evaluation, and mechanistic discussion, the work lacks the necessary originality and robust scientific foundation required for publication in Nature Communications. I recommend rejection.

Major Critiques:

Limited originality of the reaction system:

The one-pot conversion of lignin to cycloalkanes is extensively reported. The reaction design here offers no substantive breakthrough, nor does it adequately acknowledge prior literature. The claim of "surpassing the highest yields in existing literature" lacks credibility without a rigorous, transparent, and correctly described quantitative methodology.

Insufficient scientific contribution of catalyst concept:

The catalyst is essentially Ru/CNF prepared at different calcination temperatures, leading primarily to variations in Ru oxidation state. Screening calcination temperatures is a common optimization procedure. Crucially, no sufficient experimental evidence demonstrating the "dynamic Ru–RuO interface" nature of this interface is provided, leaving the core concept unsupported.

Inadequate Structural Characterization:

The authors rely solely on XPS to infer Ru⁰/Ru⁶⁺ ratios. This is insufficient. Essential complementary techniques—such as XAS (XANES/EXAFS), CO-DRIFTS, and H₂-TPR—were not employed to validate the structural model and oxidation state modulation. Drawing conclusions about interface effects and electronic structure control based on such a weak evidence base falls far short of the scientific rigor expected for a high-impact journal.

Significant Deficiencies in Data Quality and Presentation:

Image Quality: Multiple images (e.g., Fig. 1c,d) suffer from poor resolution. TEM scale bars are illegible, Ru nanoparticles are unmarked, size distribution histograms are blurred, and XRD patterns lack reference standards.

Data Validity & Methods: The stability test (Fig. 2d) was conducted under full conversion conditions, rendering the results meaningless for assessing deactivation. The yield calculation formula in SI (Section 2.4) contains fundamental errors. The basis for substrate definition (lignin vs. corncob residue?) and yield calculation is ambiguous and inconsistent.

Lack of Foundational Data: Critical evidence is missing, including: representative SEM/macroscopic images proving the CNF support structure; characterization (e.g., TGA, in-situ studies) demonstrating lignin pyrolysis occurs under the reported mild conditions (250°C, 5 MPa H₂) vs. the typical >350-500°C required; full material/carbon balances; and raw analytical data (GC, GC-MS, HPLC, NMR) supporting yield claims.

Experimental Detail & Errors: The justification for using Al₂O₃ as a control support (based solely on claimed "conductive vs. inductive" differences) is weak and unsupported by data (e.g., XPS, DRIFTS). Concerns exist about temperature probe placement/reactor filling ratio (20 mL solvent in 160 mL reactor), potentially causing systematic errors in kinetic data. Basic errors, like an inconsistent catalyst loading (0.2g vs. implied 0.1g), indicate poor manuscript preparation and review.

Detailed Concerns:

Lack of Support Structural Proof: No SEM or representative TEM images confirm the claimed CNF support structure. High-magnification, localized TEM images are insufficient and undermine the credibility of the material synthesis narrative.

Unsubstantiated Pyrolysis Conditions: The reaction conditions (250°C, 5 MPa H₂) are significantly milder than typical lignin

pyrolysis temperatures (≥ 350 - 500°C). No evidence (e.g., TGA, in-situ characterization, intermediate monitoring) is provided to prove lignin pyrolysis occurs under these conditions, invalidating the claim of "simplifying the process by integrating pyrolysis and HDO" (line 91).

Unconvincing Control Catalyst Selection: The rationale for using Al_2O_3 as a control (based on purported "conductive vs. inductive" differences) lacks depth and direct relevance to the proposed Ru-RuO mechanism. No experimental data supports the claim that support electronic properties actually differ or affect Ru's electronic state. If this comparison is central, supporting XPS/DRIFTS data is mandatory.

Poor Quality of Fig. 1c & 1d: Fig. 1c (XRD) lacks reference patterns and shows indistinguishable traces, conveying no structural information. Fig. 1d (TEM) has an illegible scale bar, unmarked Ru particles, and a blurred size distribution histogram, falling far below acceptable standards.

Fundamental Flaws in Yield Calculation & Reporting: Critical information is missing or flawed: ambiguous substrate identity (lignin vs. corncob residue), erroneous yield calculation formula (SI 2.4 - carbon vs. mass yield?), lack of a complete material/carbon balance table (essential for high-yield claims), and absence of raw analytical data supporting reported yields. All yield and performance claims are unverified and unreliable without comprehensive correction and data provision.

Reactor/Temperature Measurement Concern: The low liquid volume (20 mL in a 160 mL reactor) raises serious doubts about whether the temperature probe accurately measured the liquid phase temperature. This potential systematic error impacts all kinetic and performance data. Authors must clarify probe placement, calibration procedures, and accuracy validation.

Invalid Stability Test Methodology: Stability testing under full conversion conditions (Fig. 2d, cf. Fig. 2a) is a fundamental experimental error, as it cannot detect activity loss. These data are scientifically meaningless. Stability must be re-evaluated under quantitative (non-full conversion) conditions.

Lack of Author Diligence: The numerous errors (formulas, catalyst loading, method descriptions), lack of critical details, and poor data presentation strongly suggest inadequate internal review and a lack of responsibility, particularly from the corresponding author.

Conclusion:

Due to the profound deficiencies in novelty, mechanistic insight, experimental rigor, data quality, and presentation detailed above, the manuscript does not meet the high standards of Nature Communications. The core claims, especially regarding the "dynamic Ru-RuO interface" and superior performance, are insufficiently supported. The work requires substantial, fundamental revisions across nearly all aspects before it could be reconsidered for publication in any reputable journal. Rejection is recommended.

Reviewer #3

(Remarks to the Author)

Ma et al. reports a tunable selectivity of lignin hydrodeoxygenation (HDO) over Ru/Carbon Nanofiber (CNF) catalysts. The authors demonstrate that thermally induced restructuring of support hydroxyls generates a dynamic Ru/RuOx interface acting as surface Frustrated Lewis Pairs (sFLPs). This polarized interface facilitates heterolytic H_2 activation and phenolic adsorption, achieving a high cycloalkane yield (50% mass yield). Conversely, removing surface oxygen via high-temperature treatment shifts selectivity toward aromatics. The proposed mechanism is robustly supported by advanced characterization and theoretical calculations. This is a mechanically insightful study that establishes a clear structure-activity relationship for directing reaction pathways in biomass upgrading.

However, I have some comments that need to be properly considered before the manuscript can be accepted:

1. The characterization of the Ru/RuOx interface as 'Surface Frustrated Lewis Pairs' (sFLPs) appears to be conceptually over-packaged. As defined in the foundational literature (Stephan, Science 2016), FLP chemistry fundamentally relies on the prevention of adduct formation between Lewis acid and base sites. However, the DFT models (Fig. S21) clearly depict short Ru-O bond lengths (~ 1.7 - 1.8 \AA), indicating stable chemical ligation rather than a "frustrated" non-bonded state.

Consequently, the observed reactivity is more accurately described by established concepts such as Electronic Metal-Support Interactions or interfacial polarization. Furthermore, the manuscript lacks direct experimental evidence (e.g., atomic-resolution STEM) to confirm the specific spatial geometry required for an FLP, relying instead on theoretical inferences. The authors must rigorously justify this terminology, as the current classification seems mechanistically unjustified.

2. The product yield calculation directly employed the ratio of gas chromatographic peak areas without introducing response factors for correction, constituting a quantitative error. Furthermore, gaseous products and solid cokes haven't been well analyzed, resulting in a missing carbon balance. It is recommended that complete mass balance data be supplemented and the calculation method revised.

3. Although it is widely acknowledged that the carbon balance of biomass conversion is challenging to calculate, the authors report only liquid yields without quantifying the formation of gaseous by-products (C1-C4) or solid cokes. The absence of these data creates a significant gap in the carbon balance, potentially compromising the accuracy of product selectivity. The authors should try their best to trace all carbon atoms wherever possible, at the very least providing mass yields for gaseous products and solid cokes.

4. There is a massive pressure gap between the NAP-XPS characterization (0.2 mbar) and the actual reaction conditions (5 MPa). The surface state of Ru under 50 bar of H_2 might be significantly more reduced than what is observed at 0.2 mbar. The authors should discuss this limitation or provide evidence.

5. Regarding the potential for practical application, the authors provide real-life biomass as examples. Can authors provide scale-up to grams or hundreds of grams to demonstrate the practicality of this approach?

6. The caption for Fig. S8 references "Scheme 3," but this scheme is not present in the manuscript or supplementary information. This seems to be a typo referring either to Fig. S8 itself or a figure in the main manuscript. Please verify and correct this citation to ensure consistency.

Version 1:

Reviewer comments:

Reviewer #1

(Remarks to the Author)

The authors have taken into account all my comments and to my opinion those of other reviewers. Their manuscript is now much more robust than in their first submission. I have just now one minor revision query:

Although I appreciate the inclusion of Table S3 to position the Ru/CNF system within the current literature, the comparison remains heterogeneous and somewhat descriptive rather than rigorously quantitative. The listed studies involve different lignin types, solvents, pressures, temperatures, and product definitions, making direct comparison difficult.

Given that the manuscript claims outstanding performance, I recommend that the authors either (i) normalize the comparison where possible (e.g., clearly distinguish mass yield vs hydrocarbon yield, specify whether oxygenates are included, clarify feedstock type and severity of conditions), or (ii) more explicitly discuss the limitations of cross-study comparison and moderate any statements implying record-level performance.

Addressing this point would significantly strengthen the credibility and positioning of the work.

Reviewer #2

(Remarks to the Author)

The author well addressed all my concerns. It can be published as is.

Reviewer #3

(Remarks to the Author)

The authors properly responded to my comments and improved the manuscript in the revised manuscript. I recommend the acceptance of the manuscript for publication.

Point-by-Point Response to the Reviewer's Comments

We would like to express our sincere gratitude to the reviewers for their insightful and thorough comments on our manuscript (NCOMMS-25-98074). We also extend our appreciation to the editor for their careful handling of our submission. In response to the reviewers' comments and suggestions, we have carefully revised the manuscript, addressing each point in detail. All changes made in the revised manuscript are highlighted with a yellow background for easy of reference.

Below, we provide our point-by-point response to the reviewers' comments. We sincerely believe that these revisions have significantly strengthened the manuscript and that it now meets the journal's criteria for publication.

Detailed Response to Reviewer's Comments:

Reviewer #1:

General comment: The manuscript "Dynamic Ru/RuOx Interface on Carbon Supports Regulate Selectivity in Lignin Hydrodeoxygenation" reports a combined experimental and computational investigation of Ru catalysts supported on carbon nanofibers for lignin hydrodeoxygenation (HDO), claiming that a dynamically restructured Ru/RuOx interface governs product selectivity toward cycloalkanes versus aromatics. The work combines catalytic testing on lignin and real biomass feedstocks, near-ambient pressure XPS, and DFT/MD calculations to rationalize the observed trends. While the topic is timely and the experimental–theoretical coupling is ambitious, several key claims are currently overstated, particularly regarding the dynamic nature of the active interface and the mechanistic interpretation derived from the computational modeling. With a clearer definition of the active site, a more restrained mechanistic interpretation, and a rigorous comparison to the literature, the manuscript could become suitable for Nature Communications. I therefore recommend major revision. My main concerns are as follows:

Response: We would like to express our sincere thanks to the reviewer for the constructive and encouraging comments. Your patience, expertise, and professional insights really help us to improve the manuscript. In line with the reviewer's suggestions, we have made substantial modifications on the discussion with additional experimental and theoretical data to further support our conclusions.

We believe that the revised manuscript now fully addresses your concerns and meets the high standards of Nature Communications. We hope it is now suitable for publication.

Comment 1. A central claim of the manuscript is that a dynamic Ru/RuO_x interface regulates selectivity in lignin HDO. However, the experimental evidence primarily supports thermally induced, pretreatment-dependent interfacial states, rather than genuinely dynamic restructuring under reaction conditions. The near-ambient pressure XPS measurements are conducted under static H₂ at 300 °C and do not demonstrate reversible or reaction-driven evolution of the Ru oxidation state during catalysis. As presented, the data convincingly show controllable interface formation via CNF surface oxygen chemistry, but not dynamic interfacial behavior in the strict sense. Given the prominence of this concept in the title and abstract, the terminology should be moderated or further experimental evidence should be provided.

Response: We apologize for the ambiguity in the previous submitted version and acknowledge that the explanation was insufficiently clear. In this revised manuscript, we have carefully checked and clarified the relevant section to ensure greater precision.

It has been widely reported that oxygen-functional groups are commonly present on the surface of carbon nanofibers (CNFs). Usually, these groups are regarded as passive participants, primarily contributing to undesired byproduct formation (e.g. CO_x) during catalytic processes, and are not considered to directly modulate the active metal sites.

However, in this work, we uncover a distinct and dynamic role for surface hydroxyl (-OH) groups on CNFs. Specifically, we find that these -OH groups exhibit high surface mobility. Under reaction conditions, two adjacent hydroxyl species can migrate and undergo a recombination reaction to form adsorbed H₂O* and an oxygen adatom (2OH* → O* + H₂O*). Notably, Ru is known to catalyze this type of dihydroxylation reaction. When Ru nanoparticles are supported on CNFs, this process is significantly promoted, enabling the *in situ* generation of reactive oxygen species (O*) that strongly interact with Ru atoms to form the interfacial O-Ru-Ru motifs or heterostructure Ru/RuO_x.

This proposed mechanism is further verified by molecular dynamics simulations, which capture the mobility of surface -OH groups and their propensity to recombine in the

presence of Ru. These findings challenge the conventional view of oxygen functionalities as inert or detrimental, instead revealing their active role in constructing bifunctional Ru/RuO_x-like interfacial sites that govern catalytic performance in lignin hydrodeoxygenation reaction.

Figure R1. MD simulation of the Ru/CNF catalysts.

When CNF is treated by thermal-heating, oxygen species are partially removed, which makes OH groups deficient on the CNF surface. The restructuring reaction is mostly prohibited.

We agree with the reviewer that the active site is formed during the catalyst preparation process, it is not under the reaction conditions. The results show controllable interface formation via the oxygen species on CNF surface; it is not dynamic interfacial behavior. It is not dynamic interfacial behavior in the strict rules.

In the revised version, we have updated and changed the statement in the manuscript, including the title and the abstract on pages 1 and 2.

“Interfacial Ru/RuO_x Heterostructures on Carbon Support Regulate Selectivity in Lignin Hydrodeoxygenation”

“Constructing well-defined heterostructure interfaces in catalysts provides an approach to modulate scaling constraints and steer reaction pathways in biomass upgrading. Here, we demonstrate thermal restructuring of hydroxyl groups on carbon nanofibers (CNF) induced the formation of heterostructure of Ru/RuO_x, which function as bifunctional active sites for the one-pot hydrodeoxygenation (HDO) of lignin to liquid hydrocarbons.

The optimized 5 wt% Ru/CNF catalyst delivers outstanding performance, achieving mass/carbon yield of 49.1%/67.7%, with high selectivity toward saturated cycloalkanes. Near-ambient pressure X-ray photoelectron spectroscopy confirms that thermal treatment of CNF tunes the oxidation state of Ru. DFT calculations reveal that O-rich Ru/CNF forms interfacial heterostructure of Ru/RuO_x polarized active sites, characterized by O^{δ-}...Ru^{δ-n+}...Ru^{δ+} ensembles that heterolytically activate H₂ and strongly polarize C-O bonds in phenolic intermediates. The cooperative interplay between metallic Ru and partially oxidized RuO_x interfacial sites lowers the energy barriers for hydrogenation and deoxygenation reactions, enabling concerted reaction pathway. These insights elucidate the molecular basis of tunable selectivity in lignin HDO and demonstrate that a polarized, oxygen-decorated metal-support interface provides a general design principles for engineering next-generation catalysts for sustainable fuel production.”

Comment 2. The manuscript claims outstanding performance and, in some instances, suggests record-level yields for lignin-to-hydrocarbon conversion. Such claims are difficult to substantiate without a rigorous, quantitative comparison to the literature. I strongly recommend the inclusion of a dedicated comparison table summarizing representative HDO catalysts reported in the literature, including lignin type, reaction conditions (temperature, pressure, solvent), catalyst composition, mass yield, carbon yield, and product selectivity (aromatics vs cycloalkanes). This would greatly strengthen the manuscript by clearly positioning the Ru/CNF system relative to existing approaches and avoiding overstatement.

Response: We fully agree that a summary table would greatly assist readers in understanding our manuscript by clearly situating the Ru/CNF catalytic system within the context of other approaches reported in the literature.

Accordingly, we have compiled and compared catalytic yields from relevant studies in the supporting information. In this revised version, we have further updated the table to

include newly published literature, with the additions highlighted in yellow for easy reference.

The following Table has been updated on page 15 of the supporting information

Table S3. Summary of the converting lignin to liquid hydrocarbons from lignin and woody biomass.

Catalyst	T (°C)	P (MPa)	Feedstock	Main products	Mass/ yield	Carbon	Ref.
Ir-ReO _x /SiO ₂	260	4	Organosolv lignin	Cycloalkanes	19.3% / 29%		11 ChemSusChem 2020
			Enzymolysis lignin		9.8% / 14.6%		
			Alkaline lignin		6.6% / 9.3%		
Pt/NbOPO ₄	190	5	Birchwood	Pentanes, hexanes, alkylcyclohexanes	28.1% / -		12 Nat. Commun. 2016
Pd/m-MoO ₃ -P ₂ O ₅ /SiO ₂	180	1	Bio-oil	Pentane, hexane, methylcyclopentane, cyclohexanes	9.4% / 14.8%		13 Nat. Commun. 2017
	250	1			29.6% / 46.3%		
Ru/Nb ₂ O ₅	250	0.7	Birch lignin	Arenes, cyclohexanes	35.5% / -		14 Nat. Commun. 2017
HZSM-5	500	5	Lignin	Aromatics	- / 30%		15 Bioresour. Technol. 2015
Ni/SiO ₂ -Al ₂ O ₃	300	6	Lignin	Alkanes	42% / -		16 Chem. Commun. 2015
NiAl alloy	220	2	Poplar wood sawdust	Aromatic monomers	18.9% / -		17 Energy Fuel 2018s
Ru/Nb ₂ O ₅ -SiO ₂	230	-	Birch lignin	Arenes	19.8% / -		18 Appl. Catal. A 2017
Ni-Cu/H-Beta zeolite	330	-	Kraft lignin	Cycloalkanes	40.39% / -		19 Bioresour. Technol 2019
Ru/CNF	250	5 (at 250 °C)	corn cob	Cycloalkanes	49.1% / -		This work

Comment 3. The manuscript repeatedly attributes catalytic selectivity to surface frustrated Lewis pair (sFLP)-like ensembles involving Oδ⁻-Ruδ⁺-Ruδ⁺ motifs. While the concept is appealing, the physical identity of the active site remains insufficiently resolved. It is not clear whether the dominant functionality arises from Ru-O perimeter sites, polarized Ru-Ru ensembles, or partially oxidized Ru edge atoms. Moreover, the distinction between the proposed sFLP-like behavior and more classical metal-oxide interfacial effects is not rigorously established. At present, the sFLP terminology appears more descriptive than demonstrative, and its use should be more carefully justified or framed more cautiously.

Response: We sincerely thank the reviewer for this insightful regarding our use of the “surface frustrated Lewis pair” (sFLPs) concept. We agree that a more rigorous justification is required to distinguish our proposed $O^{\delta-}-Ru^{\delta+n+}-Ru^{\delta+}$ motif from classical metal-oxide interfacial effects. We agree that our previous description is oversimplified. The lignin conversion involves complex multiple reactions. Even for a simplified model system such as phenol hydrogenation, it involves hydrogen activation, phenol adsorption and hydrogenation, and C-O activation and cleavage. In fact, the reactions require a large ensemble of active sites instead of well-defined single or dual active sites. The hydrogen activation occurs on the $Ru^{\delta+}-Ru^{\delta+}$ partially charged pair, where $n+$ indicates a higher charge. Electronically polarized interfacial ensembles that resemble heterolytic activation motifs. The asymmetric charge distribution significantly lowered the energy barrier of H_2 activation and dissociation. The adsorption of phenol requires a large ensemble of the active sites, where the aromatic ring is adsorbed on Ru surfaces via π electron interaction (see the detailed discussion in supplementary Note 1 in the supporting information). O in the phenol is bonded with $Ru^{\delta+n+}$ site and H is associated with the $O^{\delta-}$ sites. The aromatic ring hydrogenation involves the reaction between the $H-Ru^{\delta+n+}$ and adsorbed phenol (supplementary Note 2 in the supporting information).

The following sentences have been added in the text line 440-452 of the manuscript:

“Even for a simplified model system such as phenol hydrogenation, the reaction network involves multiple interconnected steps, hydrogen activation, phenol adsorption and activation, and C–O bond cleavage, that together require a large ensemble of active sites rather than well-defined single or dual sites. Within this ensemble, sites exhibit substantial charge asymmetry and act cooperatively throughout the catalytic cycle. Hydrogen activation occurs on $Ru^{n+\delta+}-Ru^{\delta+}$ pairs embedded within a broader $O^{\delta-}-Ru^{\delta+}-Ru^{\delta+}$ ensemble, leading to heterolytic H_2 dissociation. Phenol adsorption engages a larger ensemble, where the aromatic ring interacts with Ru surfaces via π -electron interactions (Supplementary Note 1), the phenolic oxygen bonds with a $Ru^{\delta+}$ site. Subsequent ring hydrogenation involves reaction between hydridic $H-Ru^{\delta+}$ species and the adsorbed phenolic intermediate (Supplementary Note 2). These observations collectively support a model in which charge-asymmetric, oxygen-decorated Ru interfaces create a cooperative active ensemble that facilitates each elementary step in the HDO reaction network.”

In addition, in our revised manuscript, we have added significant characterization data (HAADF-STEM, XANES, EXAFS) to better resolve the physical and electronic structure of the active sites. this data confirms the co-existence of metallic Ru and partially oxidized Ru species, forming an interfacial heterostructure.

In this work, near-ambient pressure XPS was performed to verify the co-existence of Ru^0 and Ru^{n+} state on the Ru/CNF catalyst. Besides, molecular dynamic simulations were also performed to monitor the process of surface restructuring on CNF. We can clearly see that two -OH groups can migrate together and form the O^* and H_2O^* over the Ru clusters on the CNF surface. And thus, the $\text{O}^{\delta-}-\text{Ru}^{\delta+}-\text{Ru}^{\delta+}$ motifs and the interfacial Ru/ RuO_x heterostructure were formed and contributing to the active sites. Due to the difference of the electron density, the two neighboring Ru atoms are showing different chemical oxidation state. This cooperatively re-distribution produces highly polarized $\text{O}^{\delta-}-\text{Ru}^{\delta+}-\text{Ru}^{\delta+}$ pairs at the cluster-support perimeter, a defining feature of interfacial active Lewis Acid-Base Pairs. The active site is determined as the O-decorated Ru-Ru site and forming the $\text{O}^{\delta-}-\text{Ru}^{\delta+}-\text{Ru}^{\delta+}$ active site.

In accordance to the reviewer's suggestion, more characterization techniques were carried out to have better understanding on the catalyst structure. In the revised manuscript, the high-angle annular dark-field imaging (HAADF-STEM) was carried out to have much clear overview on the catalyst. From the elemental mapping of the high-resolution transmission electron microscopy (TEM) we can clearly see the Ru along the carbon nanofiber. Also, the Ru nanoparticles can also be observed. The high-angle annular dark-field STEM (HAADF-STEM) image clearly shows the obvious heterostructure between amorphous RuO_x and crystalline Ru, with distinct lattice fringes with an interplanar distance of 0.21 nm, corresponding to Ru (101) planes on the crystalline Ru structure. The results demonstrates that part of the metallic Ru is oxidized to the higher oxidation state and forming the amorphous RuO_x on the CNF, which is consistent with the near ambient pressure XPS and the DFT calculation. The restructuring of the oxygen species on CNF forms the interfacial heterostructure Ru/ RuO_x . Besides, X-ray adsorption spectroscopy (XAS) was performed to understand the active site coordination environment.

Figure R2. Illustration of the methods used for the HDO reactions. (a) Comparison of traditional method with this study for the lignin conversion, (b) Methods used in this work to synthesis carbon nanofiber, (c) HAADF-STEM images of Ru/CNF catalyst, (d) Ru K-edge XANES spectra of Ru/CNF catalysts and references, (e) FT-EXAFS spectra of Ru K-edge of Ru/CNF and Ru foil and RuO₂.

Figure R3. HADF-STEM images of Ru/CNF catalyst.

In the revised manuscript, the following sentence has been added to the discussion part in line 111-130 of the manuscript and the above figure (Figure R2) has added to Figure 1 in the manuscript on page 6, and Figure R3 has been added to Figure S2 on page 8 in the supporting information:

“The microstructure of the synthesized Ru/CNF was investigated by the high-resolution transmission electron microscopy (TEM). Energy-dispersive X-ray spectrometry (EDX) elemental mapping revealed the homogeneous distribution of Ru on the CNF. High angle annular dark-field scanning transmission electron microscopy (HAADF-STEM) image in Figure 1e shows the heterostructure of amorphous RuO_x and crystalline Ru well-defined lattice fringes with interplanar distances of 0.21 nm for the Ru (101) crystal planes. To investigate the electronic structure and coordination configuration of

Ru, X-ray absorption near-edge structure (XANES) and extended X-ray absorption fine structure (EXAFS) spectroscopy measurements were performed at the Ru K-edge, as shown in Figures 1d and 1e. In the XANES spectra, the intensity of the white line peak of Ru species in Ru/CNF is much lower than that of RuO₂ but higher than that of Ru foil. Indicating the Ru species in Ru/CNF exhibit a positive valence, and partially oxidized in a heterostructure format. Figure 1f shows the phase-corrected Fourier transform (FT) curves at the R space of the Ru/CNF in comparison with the references of Ru foil and RuO₂. Notably, Ru/CNF has two dominant peaks corresponding to Ru-O and Ru-Ru path, confirming the state of mixed metallic Ru and partially oxidized RuO_x state, consistent with the HAADF-STEM observation.”

Comment 4. The DFT and molecular dynamics calculations are technically sound and internally consistent, and they provide a coherent mechanistic narrative linking surface oxygen chemistry to hydrogen activation and C–O bond cleavage. The MD simulations capturing hydroxyl migration and partial Ru oxidation at the Ru–carbon interface are particularly convincing and qualitatively consistent with the XPS observations. The construction of OH-rich, O-rich, and OH-deficient Ru₁₈ cluster models is a reasonable approach to rationalize pretreatment effects. However, the manuscript does not sufficiently emphasize the simplified and descriptor-based nature of the computational models. The chemistry is reduced to phenol as a lignin proxy, solvent effects and high-pressure hydrogen are neglected, only a single Ru cluster size is considered, and coverage or competitive adsorption effects are not explored. These approximations are acceptable for extracting qualitative trends, but the manuscript occasionally implies a degree of quantitative correspondence with real lignin HDO that is not fully justified. The DFT results should be more explicitly framed as providing mechanistic trends and design principles rather than predictive descriptions of the full catalytic system.

Response: We agree and have revised accordingly. We are grateful for the recognition that our DFT and molecular dynamics (MD) calculations provide valuable mechanistic insights into surface restructuring.

We agree with the reviewer that our computational models are simplified ones designed to extract fundamental trends. Biomass and lignin hydrodeoxygenation (HDO) are inherently complex, involving multiphase interfaces and reaction networks. A full *ab initio* simulation of the entire system under realistic conditions remains a profound challenge. Therefore, following established practice in the field, we employed phenol as a representative model compound to probe the core chemistry of hydrogen activation, hydrogenation, and deoxygenation, key steps in lignin HDO.

Our work intentionally focuses on qualitative mechanistic trends and design principles, not on quantitative predictions for the full catalytic system. We constructed three distinct Ru₁₈ cluster models (OH-rich, O-rich, and OH-deficient) to conceptually explore how surface oxygen chemistry influences active sites. We did not aim to develop quantitative descriptors but rather to provide a qualitative narrative on how catalyst structure affects critical steps like H₂ dissociation and C-O cleavage.

For instance, our Bader charge analysis of the H₂ dissociation transition state illustrates this qualitative insight. On the Ru₁₈(OH)₈O₄ surface, the asymmetric charge distribution (Ru charges: +0.75 and +0.375) leads to shorter Ru-H bonds near oxygen and suggests a heterolytic dissociation pathway. In contrast, on the Ru₁₈(OH)₁₆ surface, the more symmetric charge distribution between Ru atoms is consistent with a homolytic dissociation pathway. This clearly shows how oxygen species modulate the electronic structure at the Ru/RuO_x interface, altering H₂ activation mechanism.

TS

TS

Figure R4. The transition state (TS) of H₂ dissociation over the Ru/CNF catalysts.

The reviewer raised a key point regarding the inherent limitations of DFT modeling studies. This insight offers valuable guidance for future research, highlighting the need to bridge the gap between idealized model catalysis and real-world catalytic conditions.

We fully agree that the role of our DFT results is to elucidate mechanistic insights and derive catalyst design principles. According, in the revised manuscript, we have modified the relevant discussion to clearly frame the computational findings in this light, ensuring they are presented as a qualitative conceptual narrative rather than a quantitative description.

A short description has been added into page 6 of the supporting information:

“Density functional theory (DFT) calculations are employed to gain fundamental insights into the role of surface oxygen chemistry in hydrogen activation and C-O bond cleavage during lignin hydrodeoxygenation (HDO). Following well-established modeling approaches in biomass conversion, phenol is used as a representative model compound, and idealized Ru₁₈ cluster models with varying oxygen coverages are constructed to explore qualitative trends. While these simplifications do not capture the full complexity of the real catalytic system, such as solvent effects, high pressure hydrogen, or competitive adsorption, and they allow us to extract key mechanistic principles and site requirements that help rationalize experimental observations and inform catalyst design.”

The following sentence has been updated on page 18 of the manuscript, and the above Figure R4 was added to Table S8 in the supporting information.

“Hydrogen adsorption and activation on the three Ru/CNF model surfaces are strongly influenced by the surface oxidation state and the extent of Ru-O coordination (Figure 5a, transition state structures in Table S7). Calculated H₂ adsorption energies increase with oxygen coverage: -0.25 eV on OH-deficient Ru₁₈(OH)₅, -0.67 eV on OH-rich Ru₁₈(OH)₁₆, and -1.01 eV on O-decorated Ru₁₈(OH)₈O₄. This trend arises from progressive electron withdrawal from surface Ru atoms toward adjacent O species, generating Ru sites with varying degrees of electron deficiency (Table S8). The resulting Ru^{δ+n+}-Ru^{δ+} pairs, while both Lewis acidic, exhibit sufficient charge asymmetry to polarize the H-H σ bond, enabling cooperative H₂ activation analogous to a Lewis acid-Lewis acid pair.”

Comment 5. The manuscript attributes selectivity primarily to differences in intrinsic reaction barriers induced by surface polarization at the Ru/RuO_x interface. While the DFT results are consistent with this interpretation, the experimental data do not fully disentangle intrinsic kinetic effects from potential influences of hydrogen availability, mass transport, or solvent-mediated stabilization. No site-normalized rates, kinetic isotope experiments, or competitive adsorption measurements are provided. As a result, the causal link between the calculated energy barriers and macroscopic selectivity remains suggestive rather than definitive and should be discussed with appropriate caution.

Response: We thank the reviewer for this insightful comment, which rightly highlights the challenge of definitively linking intrinsic kinetic barriers from DFT macroscopic selectivity in a complex, multiphase reaction system.

We fully agree that our experimental data do not isolate intrinsic kinetics from potential influences such as hydrogen availability, mass transport, or solvent effects. The absence of site-normalized rates or kinetic isotope experiments means the causal link between the calculated DFT barriers and the observed product distribution remains suggestive and interpretive, not definitive. In the revised manuscript, we have reframed our discussion to reflect this important distinction with appropriate caution.

As the reviewer notes, disentangling these factors experimentally in lignin HDO is exceptionally difficult due to the cascade of parallel and sequential reactions involving diverse, evolving intermediates. Our computational study-integrating MD simulations to model interfacial restructuring and DFT to map elementary steps is designed to provide a plausible, atomic-scale hypothesis for the observed trends. The calculations reveal how surface polarization at the Ru/RuO_x interface preferentially stabilizes transition states for key steps like C-O cleavage, offering a self-consistent electronic structure rationale for the shift in selectivity.

To strengthen the experimental foundation of this narrative, we have expanded the catalyst characterization in the revised manuscript (page 5, Figure 1c-f, Table S2) with detailed HAADF-STEM and XAS analyses. These data directly confirm the coexistence of metallic Ru and oxidized Ru species, validating the essential structural motif (the Ru/RuO_x interface) upon which our mechanistic hypothesis is built, even though it is not a quantitative manner.

In summary, while the DFT barriers provide a compelling and internally consistent mechanistic narrative, we now present them explicitly as a qualitative model that rationalizes the experimental observations, acknowledging that macroscopic selectivity emerges from a confluence of factors beyond intrinsic kinetics alone. We thank the reviewer for prompting this clearer and more nuanced discussion.

According to the reviewer's comment, we have added more advanced characterization techniques to investigate the catalytic structure and mechanistic studies in the revised manuscript, see also the above comment #3. We have revised the whole DFT discussion part in the revised manuscript.

The following discussion has been updated in line 111-130 of the manuscript:

"The physical properties of Ru/CNF catalyst pre-treated at different temperatures show similar pore structures (Table S1). X-ray diffraction patterns shows that Ru is highly dispersed on the CNF support, since no Ru species can be observed, except for the carbon species (Figure S1). The microstructure of the synthesized Ru/CNF was investigated by the high-resolution transmission electron microscopy (TEM). Energy-dispersive X-ray spectrometry (EDX) elemental mapping revealed the homogeneous distribution of Ru on the CNF. High angle annular dark-field scanning transmission

electron microscopy (HAADF-STEM) image in Figure 1c and Figure S2 shows the heterostructure of amorphous RuO_x and crystalline Ru well-defined lattice fringes with interplanar distances of 0.21 nm for the Ru (101) crystal planes. To investigate the electronic structure and coordination configuration of Ru, X-ray absorption near-edge structure (XANES) and extended X-ray absorption fine structure (EXAFS) spectroscopy measurements were performed at the Ru K-edge, as shown in Figures 1d, 1e and Table S2. In the XANES spectra, the intensity of the white line peak of Ru species in Ru/CNF is much lower than that of RuO₂ but higher than that of Ru foil. Indicating the Ru species in Ru/CNF exhibit a positive valence, and partially oxidized in a heterostructure format. Figure 1f shows the phase-corrected Fourier transform (FT) curves at the R space of the Ru/CNF in comparison with the references of Ru foil and RuO₂. Notably, Ru/CNF has two dominant peaks corresponding to Ru-O and Ru-Ru path, confirming the state of mixed metallic Ru and partially oxidized RuO_x state, consistent with the HAADF-STEM observation.”

In addition, the section in line 476-489 was modified to a conceptual framework:

“Collectively, the DFT results, using phenol as a model compound, support a mechanistic hypothesis in which surface oxygen functionalities regulate key steps within the HDO reaction network by modulating the electronic structure of the interfacial Ru/RuO_x active site. This modulation influences hydrogen activation capability and the polarization of C–O bonds in phenolic intermediates. The model suggests that an O-rich Ru/CNF surface combines electron-deficient Ru sites, which facilitate H₂ activation, with oxygenated sites that strongly bind and polarize -OH and C–O moieties. This cooperative interaction appears to lower the calculated barriers for O–H cleavage and subsequent hydrogenation steps, providing a plausible electronic-structure rationale for the enhanced HDO performance observed experimentally.

While these computational insights outline a structure-activity relationship linking surface oxidation to catalytic function-offering a framework for designing Ru-based catalysts via engineered Ru/RuO_x interfaces. Through the calculated energy landscapes describe intrinsic pathways only for the simplified model reaction, it could rationalize the structure-performance in the real lignin conversion.”

Comment 6. The manuscript is generally well written, but several sections—particularly those describing phenol activation and hydrogenation pathways—are

repetitive between the main text and the Supplementary Information. In addition, the discussion of cycloalkanes as liquid organic hydrogen carriers, while interesting, feels peripheral to the core mechanistic message and could be shortened to improve focus.

Response: We appreciate that the reviewer also agrees with that cycloalkanes as liquid hydrogen carriers are interesting.

In the field of biomass conversion and utilization, cyclohexane is usually recognized as one main component of the bio-oil. The heating value of cyclohexane is much higher than that of benzene. From the point of energy, we know that converting lignin to cycloalkanes is more meaningful than aromatics.

However, with the advantage of relatively higher hydrogen capacity (7.2%), liquid state at room temperature, high boiling points, and low toxicity, cyclohexane has been proposed and investigated as liquid hydrogen carriers (*J. Energy Chem.* 2015, 24, 587–594).

And thus, it is more significant to produce cyclohexane as the main product from lignin conversion. This is the reason why we highlight another function for cyclohexane as the liquid H₂ carrier in our work. We think it is like adding ice on the cake; by doing this the significance of cyclohexane can be even higher.

We agree with the reviewer that the discussion on liquid organic H₂ carrier should be weakened. In the revised version, we have removed the discussion on liquid organic hydrogen carrier to page 16 of the supporting information.

“This research not only provides a significant advancement in the one-pot HDO of lignin to hydrocarbons but also offers a promising approach for integrated hydrogen storage and the production of green aromatics through the selective conversion of lignin with green hydrogen to cycloalkanes, as shown in Figure S10. The use of cycloalkanes as a hydrogen carrier further highlights the potential of this process in facilitating the transportation and on-site release of hydrogen, underscoring the environmental and commercial viability of converting lignin, a byproduct of the biorefinery industry, into valuable products using Ru/CNF catalysts.”

Reviewer #2:

Overall Assessment: This manuscript presents a Ru/CNF catalytic system for the one-pot hydrodeoxygenation (HDO) of lignin into cycloalkanes, proposing a "dynamic Ru–RuO interface" to explain performance variations. While the topic—integrating renewable resource conversion, interface catalysis, and metal oxidation state modulation—is timely and relevant to sustainable chemistry and biomass upgrading, the manuscript suffers from significant shortcomings in novelty, mechanistic depth, evidential rigor, and data quality. Despite a seemingly complete framework covering catalyst synthesis, characterization, performance evaluation, and mechanistic discussion, the work lacks the necessary originality and robust scientific foundation required for publication in Nature Communications. I recommend rejection.

Response: We sincerely thank the reviewer for his/her thoughtful and rigorous evaluation of our manuscript, and insightful comments. We appreciate the recognition of the topic's relevance to sustainable chemistry and biomass valorization. At the same time, we take the concerns regarding novelty, mechanistic depth, evidential rigor, and data quality very seriously.

We appreciate the reviewer's concerns regarding novelty. In the revised manuscript, we have clarified the conceptual advance and moderated our claims to avoid overstatement. Although biomass upgrading and valorization have indeed been active areas of research for decades, they remain critically important, and highly challenging frontiers in sustainable chemistry. Continued efforts to enhance product yields, precisely tune selectivity, and improve process efficiency are essential for advancing lignocellulosic biorefineries toward economic and environmental viability. More importantly, a deep mechanistic understanding of the active sites and reaction pathways, particularly for complex, recalcitrant feedstocks like lignin is still missing. Such insights are indispensable for the rational design of next generation catalysts that deliver not only higher activity but also targeted selectivity toward high-value products.

Our study directly addresses these gaps by establishing clear structure-function relationships and uncovering the electronic and interfacial origins of catalytic performance. We believe this level of molecular-level insight represents a meaningful advance in the field and provides a generalizable framework for future catalyst development in biomass conversion and renewable fuel synthesis.

In this work, we developed a Ru catalyst supported on carbon nanofibers (Ru/CNF) and employed it for lignin hydrodeoxygenation (HDO). Under relatively mild reaction conditions, the catalyst achieves high mass yields toward liquid hydrocarbons, with performance comparing favorable to literature (Table S3 in the supporting information). Notably, the product distribution can be tuned between aromatics and cycloalkanes by modulating the catalyst surface: on the O-rich Ru/CNF surface, cycloalkanes are formed selectively with no detectable aromatics.

While thermal removal of oxygen species from carbon nanofibers surface is a well-established approach, we introduce an additional dimension by exploiting in situ surface reactions during heat treatment to deliberately tune the Ru/RuO_x interface. This strategy generates an active site ensemble with controlled charge gradients that not only facilitates lignin conversion but also enables manipulation of reaction pathways toward targeted product families. The revised manuscript clarifies the conceptual contribution in terms of interfacial electronic polarization and selectivity control.

Moreover, the nature and evolution of the active sites were systematically investigated through a synergistic combination of experimental and theoretical approaches. Molecular dynamics simulations tracked the dynamic restructuring of active sites on the CNF surface induced by in situ surface reaction during annealing, providing critical insights into their formation and stabilization of the active site ensembles.

In parallel, the key elementary steps of the hydrodeoxygenation (HDO) process, especially deoxygenation and hydrogenation were systematically elucidated through kinetic analysis, spectroscopic characterization, and DFT calculations using phenol conversion as a model reaction. It qualitatively rationalizes the structure-performance relationship for the complex lignin conversion reactions. Together, these fundamental investigations offer a robust mechanistic foundation that can inform the rational design of high-performance catalysts, enabling more precise control over activity, selectivity and stability in lignin valorization and beyond.

We believe the findings reported in our manuscript will be of strong interest to researchers working in the fields of heterogeneous catalysis and biomass conversion, as they offer fundamental insights into interfacial active site engineering, reaction mechanism, and selectivity control in lignin hydrodeoxygenation.

Below, we address the reviewer's comments in detail and describe the substantial revisions that have been implemented to further enhance the scientific rigor, clarity and impact of our manuscript. All revised text and newly added content are clearly highlighted with a yellow background for ease of identification.

Major Critiques:

Comment 1. Limited originality of the reaction system:

The one-pot conversion of lignin to cycloalkanes is extensively reported. The reaction design here offers no substantive breakthrough, nor does it adequately acknowledge prior literature. The claim of "surpassing the highest yields in existing literature" lacks credibility without a rigorous, transparent, and correctly described quantitative methodology.

Response: We sincerely thank the reviewer for the kind comments.

We agree with the reviewer that the one-pot conversion of lignin to liquid hydrocarbons has been reported in literature and remains an active area of research. However, achieving high yields of valuable products under mild conditions continues to be a significant challenge. As summarized in Table S3 in the supporting information, previous studies often require temperature as high as 350 °C to convert various biomass feedstock into liquid hydrocarbons such as alkanes and aromatics. While these reports demonstrate the feasibility of lignin conversion, our work offers two distinct and to the best of our knowledge, rarely reported advances.

First, beyond simply demonstrating catalytic activity, we show that the product distribution over Ru/CNF can be deliberately tuned between aromatics and cycloalkanes by modulating the catalyst surface chemistry. In the revised version, we have clarified the level of selectivity control.

Second, we provide fundamental insights into active sites manipulation and restructuring on the CNF surface. Carbon-based catalysts are widely used in heterogeneous catalysis, yet understanding the evolution and formation mechanism of active sites remains critical for rational catalyst design. Our findings demonstrate that oxygen species on CNF directly participate in active site formation through self-catalyzed reactions, generating asymmetric O-Ru-Ru ensembles that promote heterolytic H₂ dissociation on adjacent Ru sites. This mechanistic understanding of

active site chemistry at the molecular level is, we believe, and important contribution to the field.

Together, these advances provide fundamental understanding of active site chemistry and mechanistic insight into HDO reactions that are crucial for rational catalyst design in catalysis and biomass utilization. We trust that these novelties will be of significant interest to the readership.

In the revised version, we have updated saying in line 158-161 of the manuscript:

“To the best of our knowledge, the hydrocarbon yield reported here is among the higher values achieved to date for the conversion of lignin or woody biomass into liquid hydrocarbons, as summarized in Table S3”.

Comment 2. Insufficient scientific contribution of catalyst concept:

The catalyst is essentially Ru/CNF prepared at different calcination temperatures, leading primarily to variations in Ru oxidation state. Screening calcination temperatures is a common optimization procedure. Crucially, no sufficient experimental evidence demonstrating the "dynamic Ru–RuO interface" nature of this interface is provided, leaving the core concept unsupported.

Response: We have clarified this aspect in the revised manuscript.

We fully agree with the reviewer that thermal treatment is the most often method to tune the catalyst structure, especially used to modify the surface functional groups of the carbon-based catalyst. Here, we introduce an additional dimension by exploiting in situ surface reactions during heating treatment to deliberately tune the Ru/RuO_x interface supported on it. This strategy generates an active site ensemble with controlled charge gradients that not only facilitates lignin conversion but also enables manipulation of reaction pathways toward targeted product conversion but also enables manipulation of reaction pathways toward targeted product families. This aspect of the study highlights a potentially useful approach for controlling product distribution through rational catalyst design.

We also agree with the reviewer that the phrase “dynamic Ru-RuO_x” could be misleading, as it often implies restructuring of the interface during catalysis. In the original manuscript, this term was used to describe a distinct phenomenon: the

migration of two OH* groups on the CNF surface during annealing, which react to form O* and H₂O*, thereby modifying the local oxygen environment. This process reflects the mobility of surface oxygen species rather than dynamic changes to the Ru metal itself under reaction conditions.

To avoid ambiguity, we have revised the manuscript accordingly. The expression “dynamic Ru-RuO_x” has been replaced with the more precise description: “Interfacial Ru/RuO_x heterostructures”. This terminology better captures the static yet chemically tuned nature of the interface formed during pretreatment, which is the focus of our structure-activity analysis.

In this revised version of the manuscript, we have modified and corrected the saying as following:

“Interfacial Ru/RuO_x heterostructures”

Comment 3. Inadequate Structural Characterization:

The authors rely solely on XPS to infer Ru⁰/Ru^{δ+} ratios. This is insufficient. Essential complementary techniques—such as XAS (XANES/EXAFS), CO-DRIFTS, and H₂-TPR—were not employed to validate the structural model and oxidation state modulation. Drawing conclusions about interface effects and electronic structure control based on such a weak evidence base falls far short of the scientific rigor expected for a high-impact journal.

Response: We appreciate this important point.

In our manuscript, the catalyst structure was verified by both experimental and theoretical strategies. It has been reported that Ru can be easily oxidized to higher oxidation state, so, near-ambient pressure XPS was carried out. We found that Ru is partially oxidized to higher oxidation state, and mixed with metallic Ru. On the other hand, molecular dynamic simulation was also performed to track the restructuring process of (2OH* → O* + H₂O*). The OH-enriched carbons nanofiber surface undergoes restructuring. The hydroxyl groups possess high surface mobility and tend to migrate toward the Ru cluster edges, which is driven by the strong Ru-O affinity. The adsorbed hydroxyl groups on the Ru surface recombined together to form O* and H₂O*. The desorption of H₂O* leaves behind strongly bound O* species at the Ru-support interface. This process partially oxidizes the metallic Ru, generating O-

decorated Ru sites and yield a heterostructure of metallic Ru⁰ and Ruⁿ⁺ domains, which is consistent with the NAP-XPS discoveries. The catalyst structures were demonstrated by the combined experimental and theoretical methods.

In accordance with the reviewer's suggestion, we have added more characterizations to verify the proposed active sites in the revised manuscript.

In response to the reviewer's comment, we have performed additional high-angle annular dark-field scanning transmission electron microscopy (HAADF-STEM) and X-ray absorption spectroscopy (XANES/EXAFS) to obtain a more comprehensive overview of the catalyst structure. These techniques provide direct, element-specific insight into the local coordination environment and electronic state of Ru, which is essential for validating the interfacial Ru/RuO_x heterostructure proposed in our mechanistic model.

We note that certain complementary techniques, such as CO-DRIFTS, and H₂-TPR are not well suited for carbon-supported catalysts. In the case of CO-DRIFTS, the carbon support strongly adsorbs infrared radiation, precluding reliable detection of surface species. For H₂-TPR, the analysis is complicated by simultaneous decomposition of oxygen-containing groups on the carbon surface and surface reaction involving OH species, which obscure the reduction behavior of the Ru phase. Accordingly, we focused our characterization efforts on techniques that offer unambiguous structural information for this system.

Figure R5. Illustration of the methods used for the HDO reactions. (a) Comparison of traditional method with this study for the lignin conversion, (b) Methods used in this work to synthesis carbon nanofiber, (c) HAADF-STEM images of Ru/CNF catalyst, (d) Ru K-edge XANES spectra of Ru/CNF catalysts and references, (e) FT-EXAFS spectra of Ru K-edge of Ru/CNF and Ru foil and RuO₂.

Figure R6. HAADF-STEM images of the Ru/CNF catalyst.

In the revised manuscript, the following sentence has been added to the discussion part in line 111-130 of the manuscript and the above Figure R5 has added to Figure 1 in the manuscript on page 6, and Figure R6 has been added to Figure S2 on page 8 in the supporting information:

“The microstructure of the synthesized Ru/CNF was investigated by the high-resolution transmission electron microscopy (TEM). Energy-dispersive X-ray spectrometry (EDX) elemental mapping revealed the homogeneous distribution of Ru on the CNF. High angle annular dark-field scanning transmission electron microscopy (HAADF-STEM) image in Figure 1e shows the heterostructure of amorphous RuO_x and crystalline Ru well-defined lattice fringes with interplanar distances of 0.21 nm for the Ru (101) crystal planes. To investigate the electronic structure and coordination configuration of Ru, X-ray absorption near-edge structure (XANES) and extended X-ray absorption fine structure (EXAFS) spectroscopy measurements were performed at the Ru K-edge, as shown in Figures 1d and 1e. In the XANES spectra, the intensity of the white line peak of Ru species in Ru/CNF is much lower than that of RuO₂ but higher than that of Ru foil. Indicating the Ru species in Ru/CNF exhibit a positive valence, and partially oxidized in a heterostructure format. Figure 1f shows the phase-corrected Fourier transform (FT) curves at the R space of the Ru/CNF in comparison with the references of Ru foil and RuO₂. Notably, Ru/CNF has two dominant peaks corresponding to Ru-O and Ru-Ru path, confirming the state of mixed metallic Ru and partially oxidized RuO_x state, consistent with the HAADF-STEM observation.”

Comment 4. Significant Deficiencies in Data Quality and Presentation:

4.1. Image Quality: Multiple images (e.g., Fig. 1c, d) suffer from poor resolution. TEM scale bars are illegible, Ru nanoparticles are unmarked, size distribution histograms are blurred, and XRD patterns lack reference standards.

Response: We sincerely thank the reviewer for the kind comments.

We fully agree with the reviewer that the figures in the original version were inadequately presented and failed to meet the standard expectations for structural characterizations.

In the revised manuscript, the high-angle annular dark-field imaging (HAADF-STEM) was carried out to have much clear overview on the catalyst. We have completely reprocessed and replaced the figures with high-quality, publication-ready data. The following figures have been added and replaced the raw figures.

Figure R7. HAADF-STEM images of the Ru/CNF catalyst.

As for the XRD patterns, we also added the reference patterns (graphite). We can see that all the Ru/CNF catalysts and pure CNF are showing the carbon peak, and no Ru species diffraction peaks can be observed. The following figure has been updated as Figure S1 in the supporting information on page 7.

Figure R8. XRD patterns of the Ru/CNF catalyst with graphite.

Figure R7 and R8 has been updated on pages 7 and 8 of the supporting information.

The following sentences have been added to the text in line 111-130 of the manuscript:

“The microstructure of the synthesized Ru/CNF was investigated by the high-resolution transmission electron microscopy (TEM). Energy-dispersive X-ray spectrometry (EDX) elemental mapping revealed the homogeneous distribution of Ru on the CNF. High angle annular dark-field scanning transmission electron microscopy (HAADF-STEM) image in Figure 1e shows the heterostructure of amorphous RuO_x and crystalline Ru well-defined lattice fringes with interplanar distances of 0.21 nm for the Ru (101) crystal planes. To investigate the electronic structure and coordination configuration of Ru, X-ray absorption near-edge structure (XANES) and extended X-ray absorption fine structure (EXAFS) spectroscopy measurements were performed at the Ru K-edge, as shown in Figures 1d, 1e and Table S2. In the XANES spectra, the intensity of the white line peak of Ru species in Ru/CNF is much lower than that of RuO_2 but higher than that of Ru foil. Indicating the Ru species in Ru/CNF exhibit a positive valence, and partially oxidized in a heterostructure format. Figure 1f shows the phase-corrected Fourier transform (FT) curves at the R space of the Ru/CNF in comparison with the references of Ru foil and RuO_2 . Notably, Ru/CNF has two dominant peaks corresponding to Ru-O and Ru-Ru path, confirming the state of mixed metallic Ru and partially oxidized RuO_x state, consistent with the HAADF-STEM observation.”

4.2. Data Validity & Methods: The stability test (Fig. 2d) was conducted under full conversion conditions, rendering the results meaningless for assessing deactivation.

The yield calculation formula in SI (Section 2.4) contains fundamental errors. The basis for substrate definition (lignin vs. corncob residue?) and yield calculation is ambiguous and inconsistent.

Response: We sincerely thank the reviewer for the kind comments.

In this work, lignin is directly converted to liquid hydrocarbons via hydrodeoxygenation reaction over the Ru/CNF catalyst after the optimized reaction conditions. To evaluate the robustness of the Ru/CNF catalyst, we performed the multiple-cycle recycle tests under identical conditions and observed consistent liquid hydrocarbon yield across all the runs. Besides, the catalyst after the final cycle was subjected to XRD, XPS and HR-TEM characterizations. No obvious particle growth, and oxidation state changes were observed.

We sincerely appreciate the reviewer's insightful comment regarding the importance of conducting stability tests at relatively low conversions to better probe catalyst deactivation. For relatively simple reactions, this is indeed a well-established practice. However, the lignin conversion studied in the present work involves a complex reaction network comprising both sequential and parallel reactions, where selectivity is strongly dependent on conversion level.

In the conversion of lignin to cyclic hydrocarbons, the process proceeds via hydrogenation followed by deoxygenation. At low conversions, hydrogenation dominates while deoxygenation occurs only to a limited extent. Consequently, assessing catalyst deactivation at low conversion would not adequately capture the stability of the catalyst under relevant operating conditions where both reaction pathways are operative. Therefore, we conducted stability tests at high conversion and monitored changes in product yields, a more sensitive indicator of deactivation in this system, as it reflects the catalyst ability to sustain both hydrogenation and deoxygenation functions.

We added the following discussion in line 211-215 of the manuscript:

“To assess the catalyst stability, we performed recycling tests under semi-continuous conditions by replenishing fresh lignin feedstock without separating the catalyst, solvent, and products between cycles at the same optimized reaction conditions. The

Ru/CNF catalyst maintained consistent activity over six consecutive runs without observable deactivation (Figure S13) for both batch and continuous operation modes.”

(2) We are using the initial feedstock (corn cob residue) as the basis to calculate the yield. In the last version, we did not make it clearly and only used lignin to represent the initial feedstock. Since we only used lignin to represent the corn cob residue. It might cause misunderstanding about the calculation method. In the revised version, we have checked and modified. The mass yield of the product (C_i) was calculated by the equation:

$$\text{mass yield } (C_i, \%) = \frac{\text{mass of product } C_i}{\text{mass of feedstock input}} \times 100\%$$

The method we are using is the traditional method. The yields of liquid alkanes were determined and calculated by adding decalin as the internal standard.

We thank the reviewer again for the insightful and careful comments on the details in our manuscript. In accordance to the reviewer’s suggestion, we have corrected all the details in the manuscript.

The following sentence has been updated on page 4 in the supporting information:

“The corresponding response factors were determined by analyzing mixtures of pure, commercially obtained liquid hydrocarbons and an internal standard with precisely known masses. The yields of liquid alkanes were determined and calculated by adding decalin as the internal standard after the reaction. The mass yield of the product (C_i) was calculated by the equation, f is the relative response factor:

$$\text{mass yield } (C_i, \%) = \frac{\text{mass of } C_i}{\text{mass of feedstock input}} \times 100\%$$

$$= \frac{\text{Area } (C_i)}{\text{Area of internal standard}} \times f \times \frac{\text{mass of internal standard}}{\text{mass of feedstock input}} \times 100\%$$

The carbon yield was calculated by the equation:

$$\text{carbon yield } (C_i, \%) = \frac{\text{mass of carbon in } C_i}{\text{mass of carbon in feedstock input}} \times 100\%$$

4.3. Lack of Foundational Data: Critical evidence is missing, including: representative SEM/macrosopic images proving the CNF support structure; characterization (e.g., TGA, in-situ studies) demonstrating lignin pyrolysis occurs under the reported mild conditions (250°C, 5 MPa H₂) vs. the typical >350-500°C required; full material/carbon balances; and raw analytical data (GC, GC-MS, HPLC, NMR) supporting yield claims.

Response: We fully acknowledge that the original version of our manuscript lacked several necessary information for easy understanding. In the revised manuscript, we have addressed all the concerns.

(1) Actually, Figures S5 and S6 in the last version are showing the nano-fiber morphological structure, as displayed in the following:

Figure R11. HR-TEM images of the Ru/CNF catalyst.

Figure R12. TEM images of the carbon nanofibers.

In the revised version, to make the nano-fiber morphological structure more clear, more TEM images have been added.

In addition, we also carried out the high-angle annular dark-field imaging (HAADF-STEM) was carried out to have much clear overview on the catalyst. The nano-fiber morphological structure can also be verified.

Figure R13. Illustration of the methods used for the HDO reactions. (a) Comparison of traditional method with this study for the lignin conversion, (b) Methods used in this work to synthesis carbon nanofiber, (c) HAADF-STEM images of Ru/CNF catalyst, (d) Ru K-edge XANES spectra of Ru/CNF catalysts and references, (e) FT-EXAFS spectra of Ru K-edge of Ru/CNF and Ru foil and RuO_2 .

Figure R14. HAADF-STEM images of the Ru/CNF catalyst.

In the revised manuscript, the following sentence has been added to the discussion part in line 111-130 of the manuscript and the above Figure R13 has added to Figure 1 in the manuscript on page 6, and Figure R14 has been added to Figure S2 on page 8 in the supporting information:

“The microstructure of the synthesized Ru/CNF was investigated by the high-resolution transmission electron microscopy (TEM). Energy-dispersive X-ray spectrometry (EDX) elemental mapping revealed the homogeneous distribution of Ru on the CNF. High angle annular dark-field scanning transmission electron microscopy (HAADF-STEM) image in Figure 1e shows the heterostructure of amorphous RuO_x and crystalline Ru well-defined lattice fringes with interplanar distances of 0.21 nm for the Ru (101) crystal planes. To investigate the electronic structure and coordination configuration of Ru, X-ray absorption near-edge structure (XANES) and extended X-ray absorption fine

structure (EXAFS) spectroscopy measurements were performed at the Ru K-edge, as shown in Figures 1d and 1e. In the XANES spectra, the intensity of the white line peak of Ru species in Ru/CNF is much lower than that of RuO₂ but higher than that of Ru foil. Indicating the Ru species in Ru/CNF exhibit a positive valence, and partially oxidized in a heterostructure format. Figure 1f shows the phase-corrected Fourier transform (FT) curves at the R space of the Ru/CNF in comparison with the references of Ru foil and RuO₂. Notably, Ru/CNF has two dominant peaks corresponding to Ru-O and Ru-Ru path, confirming the state of mixed metallic Ru and partially oxidized RuO_x state, consistent with the HAADF-STEM observation.”

(2) We agree that our original phrasing-particularly the term “pyrolysis”-was misleading in the context of the mild reaction conditions employed (5 MPa H₂ at 250 °C). Classical thermal pyrolysis of lignin typically requires temperature higher than 350 °C. However, it is not the same case in our manuscript.

In fact, our system operates via catalytic hydrodeoxygenation, not conventional pyrolysis. The Ru/CNF catalyst facilitates direct cleavage of β-O-4 and other ether linkages in lignin under H₂ at relatively low temperatures, a process well-documented in recent literature. Under these conditions, lignin depolymerization proceeds through hydrogen-assisted C-O bond scission rather than thermal cracking. And thus, a relatively lower temperature is needed.

We have removed the term “pyrolysis” and clarified that the process proceeds via catalytic depolymerization under hydrogen. The following sentence was updated in line 96-99 of the manuscript:

“Another strategy employed in this work is the one-pot catalytic conversion process, which streamlines lignin valorization by integrating depolymerization and hydrodeoxygenation (HDO) into a single step. In this study, we focus on this one-pot approach to directly convert lignin into value-added liquid hydrocarbons.”

(3) we appreciate the reviewer’s comments regarding the inherent challenges in establishing a complete carbon balance in biomass conversion processes, which involve gaseous, liquid and solid product fractions.

In our study, the reaction was conducted under a higher H₂ pressure (5 MPa). We did attempt to quantify gaseous products by GC-FID, however, only trace amounts of

methane, originating from minor C-C bond cleavage in lignin side chains were detected. The concentration was extremely low (below micro mol per gram of lignin), and due to the severe dilution in the large excess amount of H₂, accurate quantification was not feasible. Consequently, gaseous products were not considered from the mass yield calculation, as their contribution is negligible relative to the liquid phase products.

For the liquid phase, products were separated from the solid residue via filtration, and quantified by GC-MS and GC-FID using decalin as an internal standard to ensure accuracy and reproducibility.

The solid fraction, comprising unconverted carbonaceous residue and the catalyst was collected, thoroughly washed, dried and weighted. The net solid residue mass was determined by subtracting the catalyst. While, mass losses during handling or incomplete recovery cannot entirely avoided (also commonly reported in the literature). And thus, the solid yield is rarely reported, and liquid phase products are the main focus in the literature, for example: *Nat. Commun.* 2017, 8, 16104); *Nat. Commun.* 2017, 8, 591, *Chem* 2019, 5, 1-13, etc.

Taken together, the reported carbon and mass yields are based on quantified liquid products, with gaseous contributions deemed insignificant and thus omitted for simplicity and clarity. This methodology aligns with common practice in lignin HDO studies where gas-phase products are minimal under hydrogen-rich conditions.

In the revised version, we have made it clear in the experimental part on page 4 of the supporting information.

“The corresponding response factors were determined by analyzing mixtures of pure, commercially obtained liquid hydrocarbons and an internal standard with precisely known masses. The yields of liquid alkanes were determined and calculated by adding decalin as the internal standard after the reaction.”

4.4. Experimental Detail & Errors: The justification for using Al₂O₃ as a control support (based solely on claimed "conductive vs. inductive" differences) is weak and unsupported by data (e.g., XPS, DRIFTS). Concerns exist about temperature probe placement/reactor filling ratio (20 mL solvent in 160 mL reactor), potentially causing

systematic errors in kinetic data. Basic errors, like an inconsistent catalyst loading (0.2g vs. implied 0.1g), indicate poor manuscript preparation and review.

Response: We sincerely thank the reviewer for the kind comments. We agree with the reviewer that the discussion on conductive and inductive is weak. They are not sufficiently tied to the proposed Ru-RuO₂ interfacial mechanism.

Alumina and activated carbon (AC) are two important and typical supports that are used in multiple catalytic systems. That's why we choose the commercial Al₂O₃ and AC as the benchmark catalysts in our manuscript.

In the revised version, we have re-framed the discussion, and removed the content on “conductive” and “inductive”.

In addition, we fully agree with the reviewer that the relatively low liquid volume in a 160 ml reactor could raise questions about the measurement and test, particularly if the probe was not within the reaction mixture.

In our experiments, a 160 mL Parr reactor was used. The reactor is equipped with two probes extending from the head into the vessel: a stirring shaft with impeller and a thermocouple well containing the temperature sensor. Both the stirring and the thermocouple tip was fully immersed in the liquid phase throughout the reaction. A consistent stirring rate was applied in all experiments to ensure vigorous mixing and to minimize thermal gradients between the liquid, headspace and reactor walls.

To verify that the thermocouple accurately measured the reaction temperature, we conducted preliminary test using distilled water to determine the minimum allowable liquid volume for reliable operation. Under the conditions employed, we are confident that the measured temperature reflects that of the reaction solvent, and that vigorous mixing effectively minimized any thermal gradients.

In the revised version, we have added the following sentences to the experimental part in the supporting information on page 3.

We have modified the error in the experimental, in this work, 0.1 g lignin and 0.1 catalyst was used for the HDO reaction.

The following sentences has been updated on page 4 of the supporting information

“The direct hydrodeoxygenation of lignin was performed in a 160 ml stainless Hastelloy autoclave (Parr reactor). In a typical run, feedstock (lignin 0.1 g), catalyst (0.1 g), and dodecane (20 ml) were loaded into the autoclave and sealed. Herein, both the stirring bar and the sensing tip of the thermal couple are fully immersed in the solvent.”

Comment 5. Detailed Concerns:

5.1. Lack of Support Structural Proof: No SEM or representative TEM images confirm the claimed CNF support structure. High-magnification, localized TEM images are insufficient and undermine the credibility of the material synthesis narrative.

Response: We sincerely thank the reviewer for the kind comments.

We agree with the reviewer that clear morphological evidence of the carbon nanofibers support structure is essential to support the discussion in our manuscript. Actually, the following figure in the last version are showing the nano-fiber morphological structure, as displayed in the following Figures.

Figure R15. HR-TEM images of the Ru/CNF catalyst.

In the revised version, to make the nano-fiber morphologic more clear, the following figures are added into the supporting information.

Figure R16. TEM images of the carbon nanofibers.

We also include the energy-dispersive X-ray spectroscopy (EDS) mapping to further confirm the spatial distribution of Ru along the nano-fiber. From all these TEM images, we think the nano-fiber morphologic structure can be clearly proved. The expanded dataset now provides a more comprehensive and convincing structural basis for our material synthesis claims.

Figure R17. EDS and elemental mapping of Ru/CNF catalyst.

On the other hand, our group used this method to prepare carbon nanofibers and have been used in multiple catalytic reactions (*J. Catal.*, 2005, 229, 82-96; *Carbon*, 2007, 45, 785-796; *ACS Catal.*, 2024, 6, 4139-4154).

In accordance to the reviewer's suggestion, more characterization techniques were carried out to have better understanding on the catalyst structure. In the revised manuscript, the high-angle annular dark-field imaging (HAADF-STEM) was carried

out to have much clear overview on the catalyst. From the elemental mapping of the high-resolution transmission electron microscopy (TEM) we can clearly see the Ru along the carbon nanofiber. Also, the Ru nanoparticles can also be observed. The high-angle annular dark-field STEM (HAADF-STEM) image clearly shows the obvious heterostructure between amorphous RuO_x and crystalline Ru, with distinct lattice fringes with an interplanar distance of 0.21 nm, corresponding to Ru (101) planes on the crystalline Ru structure. The results demonstrates that part of the metallic Ru is oxidized to the higher oxidation state and forming the amorphous RuO_x on the CNF, which is consistent with the near ambient pressure XPS and the DFT calculation. The restructuring of the oxygen species on CNF forms the interfacial heterostructure Ru/ RuO_x . Besides, X-ray adsorption spectroscopy (XAS) was performed to understand the active site coordination environment.

Figure R18. Illustration of the methods used for the HDO reactions. (a) Comparison of traditional method with this study for the lignin conversion, (b) Methods used in this work to synthesis carbon nanofiber, (c) HAADF-STEM images of Ru/CNF catalyst, (d) Ru K-edge XANES spectra of Ru/CNF catalysts and references, (e) FT-EXAFS spectra of Ru K-edge of Ru/CNF and Ru foil and RuO_2 .

Figure R19. HADF-STEM images of Ru/CNF catalyst.

In the revised manuscript, the following sentence has been added to the discussion part in line 111-130 of the manuscript and the above figure (Figure R18) has added to Figure 1 in the manuscript on page 6, and Figure R19 has been added to Figure S2 on page 8 in the supporting information:

“The microstructure of the synthesized Ru/CNF was investigated by the high-resolution transmission electron microscopy (TEM). Energy-dispersive X-ray spectrometry (EDX) elemental mapping revealed the homogeneous distribution of Ru on the CNF. High angle annular dark-field scanning transmission electron microscopy (HAADF-STEM) image in Figure 1e shows the heterostructure of amorphous RuO_x and crystalline Ru well-defined lattice fringes with interplanar distances of 0.21 nm for the Ru (101) crystal planes. To investigate the electronic structure and coordination configuration of

Ru, X-ray absorption near-edge structure (XANES) and extended X-ray absorption fine structure (EXAFS) spectroscopy measurements were performed at the Ru K-edge, as shown in Figures 1d and 1e. In the XANES spectra, the intensity of the white line peak of Ru species in Ru/CNF is much lower than that of RuO₂ but higher than that of Ru foil. Indicating the Ru species in Ru/CNF exhibit a positive valence, and partially oxidized in a heterostructure format. Figure 1f shows the phase-corrected Fourier transform (FT) curves at the R space of the Ru/CNF in comparison with the references of Ru foil and RuO₂. Notably, Ru/CNF has two dominant peaks corresponding to Ru-O and Ru-Ru path, confirming the state of mixed metallic Ru and partially oxidized RuO_x state, consistent with the HAADF-STEM observation.”

5.2. Unsubstantiated Pyrolysis Conditions: The reaction conditions (250°C, 5 MPa H₂) are significantly milder than typical lignin pyrolysis temperatures (≥350-500°C). No evidence (e.g., TGA, in-situ characterization, intermediate monitoring) is provided to prove lignin pyrolysis occurs under these conditions, invalidating the claim of "simplifying the process by integrating pyrolysis and HDO" (line 91).

Response: We sincerely thank the reviewer for the kind comments.

We agree that our original phrasing-particularly the term “pyrolysis”-was misleading in the context of the mild reaction conditions employed (5 MPa H₂ at 250 °C). Classical thermal pyrolysis of lignin typically requires temperature higher than 350 °C. However, it is not the same case in our manuscript.

In fact, our system operates via catalytic hydrodeoxygenation, not conventional pyrolysis. The Ru/CNF catalyst facilitates direct cleavage of β-O-4 and other ether linkages in lignin under H₂ at relatively low temperatures, a process well-documented in recent literature. Under these conditions, lignin depolymerization proceeds through hydrogen-assisted C-O bond scission rather than thermal cracking. And thus, a relatively lower temperature is needed.

Furthermore, there are some literatures reporting that lignin or biomass can be converted to liquid hydrocarbons at the temperatures lower than 300 °C (*Nat. Commun.*

2017, 8, 591; *Nat. Commun.* 2017, 8, 16104; *ChemSusChem* 2020, 13, 4409-4419). Different with the traditional two-step method, this one-pot process can convert raw lignin or biomass directly to liquid hydrocarbons with the assistance of H₂. This is becoming an accepted paradigm.

To make it clear about the reaction process, we have removed the term “pyrolysis” from the revised manuscript and replaced it with catalytic depolymerization. **The following sentence was updated in line 96-99 of the manuscript.**

“Another strategy employed in this work is the one-pot catalytic conversion process, which streamlines lignin valorization by integrating depolymerization and hydrodeoxygenation (HDO) into a single step. In this study, we focus on this one-pot approach to directly convert lignin-rich corncob into value-added liquid hydrocarbons.”

5.3. Unconvincing Control Catalyst Selection: The rationale for using Al₂O₃ as a control (based on purported "conductive vs. inductive" differences) lacks depth and direct relevance to the proposed Ru-RuO mechanism. No experimental data supports the claim that support electronic properties actually differ or affect Ru's electronic state. If this comparison is central, supporting XPS/DRIFTS data is mandatory.

Response: We sincerely thank the reviewer for the kind comments.

Alumina (Al₂O₃) and activated carbon (AC) were selected as benchmark supports because they represent two widely used catalyst classes in heterogeneous catalysis and allow us to contextualize the performance of our Ru/CNF catalyst. However, the more critical rationale, and one that directly relates to our mechanistic hypothesis lies in the distinct behavior of surface hydroxyl groups on these supports.

On alumina, hydroxyl groups are strongly adsorbed and generally considered immobile under thermal treatment conditions (*Fuel*, 2025, 379, 133066). This precludes significant surface restructuring for Ru supported on Al₂O₃. In contrast, hydroxyl groups on carbon nanofibers (CNF) exhibit sufficient surface mobility to enable dynamic restructuring during annealing, leading to the formation of the interfacial Ru/RuO_x heterostructure that is central of our proposed mechanism. Thus, comparing Ru/Al₂O₃ and Ru/AC with Ru/CNF allows us to isolate the effect of support-induced surface restructuring or the lack thereof on catalytic performance.

It is important to clarify that our study does not aim to systematically investigate support electronic effects through direct comparison with Ru/Al₂O₃ or Ru/AC. Rather, these benchmarks are included to evaluate the catalytic performance of Ru/CNF for lignin conversion to liquid hydrocarbons relative to established catalyst systems. The mechanistic insights into active site evolution are derived from the oxygen-mediated restructuring on the CNF surface itself, as supported by our experimental and computational analyses.

In the revised manuscript, we have modified the discussion in line 169-176 of the manuscript:

“Ruthenium supported catalysts have demonstrated high activity in the HDO of the lignin model compounds. For comparison, we selected Ru/Al₂O₃ and Ru/AC catalysts as benchmark catalysts. The Ru/AC catalyst exhibited slightly higher yields than Ru/Al₂O₃, with the mass and carbon yields of 14.8% and 21.6%, respectively. Importantly, the choice of these benchmarks also reflects differences in surface hydroxyl mobility: while hydroxyl groups on Al₂O₃ are strongly bound and immobile, limiting surface restructuring, the mobile hydroxyl species on CNFs enable the formation of the interfacial Ru/RuO_x heterostructure under thermal treatment, which we identify as key to the enhanced HDO performance.”

5.4. Poor Quality of Fig. 1c & 1d: Fig. 1c (XRD) lacks reference patterns and shows indistinguishable traces, conveying no structural information. Fig. 1d (TEM) has an illegible scale bar, unmarked Ru particles, and a blurred size distribution histogram, falling far below acceptable standards.

Response: We sincerely thank the reviewer for the kind comments.

We fully agree with the reviewer that the figures in the original version were inadequately presented, we have carefully re-plotted the figures in the revised version.

The reference standard has been added to the XRD patterns, as shown on page 8 of the supporting information.

Figure R20. XRD patterns of the Ru/CNF catalyst with graphite.

In the revised manuscript, the high-angle annular dark-field imaging (HAADF-STEM) was carried out to have much clear overview on the catalyst. We have completely reprocessed and replaced the figures with high-quality, publication-ready data. The following figures have been added and replaced the raw figures.

From the elemental mapping of the high-resolution transmission electron microscopy (TEM) we can clearly see the Ru along the carbon nanofiber. Also, the Ru nanoparticles can also be observed. The high-angle annular dark-field STM (HAADF-STEM) image clearly shows the obvious heterostructure between amorphous RuO_x and crystalline Ru, with distinct lattice fringes with an interplanar distance of 0.21 nm, corresponding to Ru (101) planes on the crystalline Ru structure. The results demonstrates that part of the metallic Ru is oxidized to the higher oxidation state and forming the amorphous RuO_x on the CNF, which is consistent with the near ambient pressure XPS and the DFT calculation. The restructuring of the oxygen species on CNF forms the interfacial heterostructure Ru/ RuO_x .

Figure R21. HAADF-STEM images of the Ru/CNF catalyst.

The following sentences have been added to the main text in line 111-130 of the manuscript:

“The microstructure of the synthesized Ru/CNF was investigated by the high-resolution transmission electron microscopy (TEM). Energy-dispersive X-ray spectrometry (EDX) elemental mapping revealed the homogeneous distribution of Ru on the CNF. High angle annular dark-field scanning transmission electron microscopy (HAADF-STEM) image in Figure 1e shows the heterostructure of amorphous RuO_x and crystalline Ru well-defined lattice fringes with interplanar distances of 0.21 nm for the Ru (101) crystal planes.”

5.5. Fundamental Flaws in Yield Calculation & Reporting: Critical information is missing or flawed: ambiguous substrate identity (lignin vs. corncob residue), erroneous yield calculation formula (SI 2.4 - carbon vs. mass yield?), lack of a complete

material/carbon balance table (essential for high-yield claims), and absence of raw analytical data supporting reported yields. All yield and performance claims are unverified and unreliable without comprehensive correction and data provision.

Response: We sincerely thank the reviewer for the kind comments.

We are using the initial feedstock (corn cob residue) as the basis to calculate the yield. In the last version, we did not make it clearly and only used lignin to represent the initial feedstock. Since we only used lignin to represent the corn cob residue. It might cause misunderstanding about the calculation method. In the revised version, we have checked and modified. The mass yield of the product (C_i) was calculated by the equation:

$$\text{mass yield } (C_i, \%) = \frac{\text{mass of product } C_i}{\text{mass of feedstock input}} \times 100\%$$

The method we are using is the traditional method. The yields of liquid alkanes were determined and calculated by adding decalin as the internal standard.

We thank the reviewer again for the insightful and careful comments on the details in our manuscript. In accordance to the reviewer's suggestion, we have corrected all the details in the manuscript.

The following sentence has been updated on page 4 in the supporting information

“The corresponding response factors were determined by analyzing mixtures of pure, commercially obtained liquid hydrocarbons and an internal standard with precisely known masses. The yields of liquid alkanes were determined and calculated by adding decalin as the internal standard after the reaction. The mass yield of the product (C_i) was calculated by the equation, f is the relative response factor:

$$\text{mass yield } (C_i, \%) = \frac{\text{mass of } C_i}{\text{mass of feedstock input}} \times 100\%$$

$$= \frac{\text{Area } (C_i)}{\text{Area of internal standard}} \times f \times \frac{\text{mass of internal standard}}{\text{mass of feedstock input}} \times 100\%$$

The carbon yield was calculated by the equation:

$$\text{carbon yield (C}_i\text{, \%)} = \frac{\text{mass of carbon in C}_i}{\text{mass of carbon in feedstock input}} \times 100\%$$

5.6. Reactor/Temperature Measurement Concern: The low liquid volume (20 mL in a 160 mL reactor) raises serious doubts about whether the temperature probe accurately measured the liquid phase temperature. This potential systematic error impacts all kinetic and performance data. Authors must clarify probe placement, calibration procedures, and accuracy validation.

Response: We appreciate this insightful and careful comment, which highlights a critical aspect of our experimental setup.

We fully agree with the reviewer that the relatively low liquid volume in a 160 ml reactor could raise questions about the measurement and test, particularly if the probe was not within the reaction mixture.

In our experiments, we used the Parr reactor with the volume of 160 ml. There are two long bars (or probes) from the top to the bottom of the reactor, one is the stirring bar, and the other one is the thermal couple bar. The stirring bar and the sensing tip of the thermal couple are fully immersed in the liquid phase. Every time the same stirring rate was used, which ensured vigorous mixing and minimized the thermal gradients between the liquid, headspace and reactor walls. We also used distilled water to measure the minimum volume can be used. Therefore, we think the temperature is measuring the reaction solvent, and with the help of vigorous mixing, the thermal gradients were minimized.

In the revised version, we have added the following sentences to the experimental part in the supporting information on page 4.

“The direct hydrodeoxygenation of lignin was performed in a 160 ml stainless Hastelloy autoclave (Parr reactor). In a typical run, feedstock (lignin 0.1 g), catalyst (0.1 g), and dodecane (20 ml) were loaded into the autoclave and sealed. Herein, both the stirring bar and the sensing tip of the thermal couple are fully immersed in the solvent.”

5.7. Invalid Stability Test Methodology: Stability testing under full conversion conditions (Fig. 2d, cf. Fig. 2a) is a fundamental experimental error, as it cannot detect

activity loss. These data are scientifically meaningless. Stability must be re-evaluated under quantitative (non-full conversion) conditions.

Response: We sincerely thank the reviewer for this insightful and kind comment.

Please see the details in the answer to comment 4.2.

5.8. Lack of Author Diligence: The numerous errors (formulas, catalyst loading, method descriptions), lack of critical details, and poor data presentation strongly suggest inadequate internal review and a lack of responsibility, particularly from the corresponding author.

Response: We first thank the reviewer for the kind comments.

We sincerely apologize for the errors, mistakes or omissions in the original manuscript, including inaccuracies in formulae, insufficient experimental details. These shortcomings should not appear in the submission.

The concerns raised by the reviewer have promoted us to implement a more stringent internal validation for all the details listed in the manuscript and supporting information. We have cross-verified all chemical formulae, reaction conditions, and numerical values across raw data and the manuscript. We also added missing methodological details. The unclear figures have been updated with improved clarity, proper labeling and statistical annotations. See the related modification in the whole manuscript and supporting information.

We deeply regret that the initial version did not meet the expected standards of Nature Communications. After careful and substantial revision, we believe the quality of our manuscript has been improved.

Conclusion:

Due to the profound deficiencies in novelty, mechanistic insight, experimental rigor, data quality, and presentation detailed above, the manuscript does not meet the high standards of Nature Communications. The core claims, especially regarding the "dynamic Ru-RuO interface" and superior performance, are insufficiently supported. The work requires substantial, fundamental revisions across nearly all aspects before it could be reconsidered for publication in any reputable journal. Rejection is recommended.

Response: We sincerely thank the reviewer for the thoughtful evaluation of our manuscript. We acknowledge the seriousness of the concerns raised regarding the novelty, mechanistic insights, experimental rigor, data quality and presentation.

We take the reviewer's comments seriously and appreciate the detailed feedback provided. Upon careful reflection, we recognized that certain aspects of our work and its contextualization could have been presented with greater clarity, depth, and rigor. The reviewer's insights will be invaluable as we work to strengthen the manuscript.

Although the submitted version fell short of the high standards of *Nature Communications*, we believe the core scientific question-efficient catalytic valorization of lignin via tunable metal-oxide interfaces remains significant and of interest to the broader catalysis and biomass conversion communities. With substantial revisions addressing the concerns raised, we are committed to improving the manuscript to meet the expectations of a reputable journal.

We appreciate the opportunity to have submitted our work to *Nature Communications* and are grateful for the constructive comments, which will undoubtedly improve the quality of this research. We hope that, after significant revision, we may be able to substantially strengthened the quality of our manuscript for revision.

Reviewer #3:

General comment: Ma et al. reports a tunable selectivity of lignin hydrodeoxygenation (HDO) over Ru/Carbon Nanofiber (CNF) catalysts. The authors demonstrate that thermally induced restructuring of support hydroxyls generates a dynamic Ru/RuOx interface acting as surface Frustrated Lewis Pairs (sFLPs). This polarized interface facilitates heterolytic H₂ activation and phenolic adsorption, achieving a high cycloalkane yield (50% mass yield). Conversely, removing surface oxygen via high-temperature treatment shifts selectivity toward aromatics. The proposed mechanism is robustly supported by advanced characterization and theoretical calculations. This is a mechanically insightful study that establishes a clear structure-activity relationship for directing reaction pathways in biomass upgrading.

However, I have some comments that need to be properly considered before the manuscript can be accepted:

Response: We sincerely thank the reviewer for his/her constructive and insightful comments, which have significantly helped us improve the quality and clarity of our manuscript. In response to the reviewer's suggestions, we have provided the requested experimental data to further substantiate the key concepts presented in this work. We believe these updated results have strengthened the scientific rigor and overall impact of our study.

We are confident that the revisions address the reviewer's concerns and meet the journal's high standards. We therefore believe the manuscript is now well-suited for publication in Nature Communication. Below, we will address the reviewer's comments point by point. All revisions have been clearly highlighted in the revised manuscript and supplementary information files for ease of reference.

Comment 1. The characterization of the Ru/RuOx interface as 'Surface Frustrated Lewis Pairs' (sFLPs) appears to be conceptually over-packaged. As defined in the foundational literature (Stephan, Science 2016), FLP chemistry fundamentally relies on the prevention of adduct formation between Lewis acid and base sites. However, the DFT models (Fig. S21) clearly depict short Ru-O bond lengths (~1.7-1.8 Å), indicating stable chemical ligation rather than a "frustrated" non-bonded state. Consequently, the observed reactivity is more accurately described by established concepts such as

Electronic Metal-Support Interactions or interfacial polarization. Furthermore, the manuscript lacks direct experimental evidence (e.g., atomic-resolution STEM) to confirm the specific spatial geometry required for an FLP, relying instead on theoretical inferences. The authors must rigorously justify this terminology, as the current classification seems mechanistically unjustified.

Response: We sincerely express our great thanks for the reviewer's insightful and constructive comment, which has helped us refine the conceptual framing of our work. The reviewer correctly notes that, as defined in the foundational sFLP literature (Stephan, Science 2016), frustrated Lewis pairs fundamentally rely on steric prevention of dative bond formation between Lewis acid and base sites. We agree that the Ru-O bond lengths observed in our DFT models (~ 1.7 - 1.8 Å) indicate stable chemical ligation, which would indeed be inconsistent with a strictly "frustrated" non-bonded state as originally defined. We note that the concept of solid surface frustrated Lewis pairs (sFLPs) has evolved in recent years to encompass systems where Lewis Acid-Base pairs exhibit cooperative reactivity despite the presence of chemical bonding. For example: Tsang and co-worker reported a Co-N sFLP system with a bond length of 1.6 Å (*J. Am. Chem. Soc.* 2021, 143, 21294–21301). The same group described a Ru-O sFLP system with a Ru-O bond length of 1.9 Å (*J. Am. Chem. Soc.* 2023, 145, 14548-14561).

The Ru-O distances in our work are comparable to or slightly longer than these literature values, suggesting that our system shares features with previously reported sFLP systems where bond formation and cooperative reactivity coexist.

Nevertheless, we agree with the reviewer that this terminology should be applied with caution and rigorous justification. To avoid any potential conceptual overreach or misinterpretation. We have removed the "frustrated Lewis pair" framing from the revised manuscript. In its place, we now describe the active site using the more precise and empirically grounded term "interfacial Ru/RuO_x heterostructure". This terminology accurately reflects the structural and electronic features revealed by our combined experimental and computational analyses without invoking a specific mechanistic label that may not be fully justified. Corresponding revisions have been made throughout the manuscript.

Importantly, while we have stepped back from the sFLPs terminology, the underlying concept of a cooperative Lewis acid-base pair at the Ru/RuO_x interface remains strongly supported by our data:

Near-ambient pressure XPS confirms the co-existence of metallic Ru⁰ and oxidized Ruⁿ⁺ species, establishing the mixed-valence character of the interface. HAADF-STEM (newly added in the revised manuscript) directly visualizes the heterostructure with distinct lattice fringes for Ru and amorphous RuO_x domains. XANES and EXAFS analysis provides quantitative evidence for the local coordination environment and electronic structure of Ru. Molecular dynamic simulations track the restructuring process (2OH * → O * + H₂O *) and reveal how mobile surface oxygen species on CNF facilitate formation of the asymmetric Ru-O-Ru ensemble (Figure 4). Notably, Ru itself has been reported to catalyze such restructuring process (*Javier Pérez-Ramírez, et al., J. Catal., 2008, 255, 29-39*), suggesting a synergistic effect that promotes formation of the active interfacial structure.

Thus, while we have refined the conceptual framing in response to the reviewer's critique, the core mechanistic insight, that an asymmetric, charge-polarized Ru/RuO_x interface enables cooperative activation of H₂ and phenolic intermediates, remains robustly supported by both experimental and theory. We are grateful to the reviewer for promoting this more precise and carefully considered presentation.

Figure R22. Molecular dynamic simulation of the Ru/CNF catalysts. (a, c) Ru supported on the OH-deficient CNF surface, (b, d) Ru supported on the O-rich CNF surface.

In accordance to the reviewer's suggestion, and to have better understanding on the catalyst structure, we have added more characterizations, the atomic-resolution STEM and X-ray absorption spectroscopy (XAS) in the revised version, also shown in the following Figures.

From the elemental mapping of the high-resolution transmission electron microscopy (TEM) we can clearly see the Ru along the carbon nanofiber. Also, the Ru nanoparticles can also be observed. The high-angle annular dark-field STEM (HAADF-STEM) image clearly shows the obvious heterostructure between amorphous RuO_x and crystalline Ru, with distinct lattice fringes with an interplanar distance of 0.21 nm, corresponding to Ru (101) planes on the crystalline Ru structure. The results demonstrates that part of the metallic Ru is oxidized to the higher oxidation state and forming the amorphous RuO_x on the CNF, which is consistent with the near ambient pressure XPS and the DFT calculation. The restructuring of the oxygen species on CNF forms the interfacial heterostructure Ru/ RuO_x .

Figure R23. Illustration of the methods used for the HDO reactions. (a) Comparison of traditional method with this study for the lignin conversion, (b) Methods used in this work to synthesis carbon nanofiber, (c) HAADF-STEM images of Ru/CNF catalyst, (d) Ru K-edge XANES spectra of Ru/CNF catalysts and references, (e) FT-EXAFS spectra of Ru K-edge of Ru/CNF and Ru foil and RuO_2 .

Figure R24. Atomic-resolution STEM images of the Ru/CNF catalyst.

In the revised manuscript, the following sentence has been added to the discussion part in line 111-130 of the manuscript and the above figure (Figure R23) has added to Figure 1 in the manuscript on page 6, and Figure R24 has been added to Figure S2 on page 8 in the supporting information:

“The microstructure of the synthesized Ru/CNF was investigated by the high-resolution transmission electron microscopy (TEM). Energy-dispersive X-ray spectrometry (EDX) elemental mapping revealed the homogeneous distribution of Ru on the CNF. High angle annular dark-field scanning transmission electron microscopy (HAADF-STEM) image in Figure 1e shows the heterostructure of amorphous RuO_x and crystalline Ru well-defined lattice fringes with interplanar distances of 0.21 nm for the Ru (101) crystal planes. To investigate the electronic structure and coordination configuration of

Ru, X-ray absorption near-edge structure (XANES) and extended X-ray absorption fine structure (EXAFS) spectroscopy measurements were performed at the Ru K-edge, as shown in Figures 1d and 1e. In the XANES spectra, the intensity of the white line peak of Ru species in Ru/CNF is much lower than that of RuO₂ but higher than that of Ru foil. Indicating the Ru species in Ru/CNF exhibit a positive valence, and partially oxidized in a heterostructure format. Figure 1f shows the phase-corrected Fourier transform (FT) curves at the R space of the Ru/CNF in comparison with the references of Ru foil and RuO₂. Notably, Ru/CNF has two dominant peaks corresponding to Ru-O and Ru-Ru path, confirming the state of mixed metallic Ru and partially oxidized RuO_x state, consistent with the HAADF-STEM observation.”

Comment 2. The product yield calculation directly employed the ratio of gas chromatographic peak areas without introducing response factors for correction, constituting a quantitative error. Furthermore, gaseous products and solid cokes haven't been well analyzed, resulting in a missing carbon balance. It is recommended that complete mass balance data be supplemented and the calculation method revised.

Response: We thank the reviewer for your insightful comments. As the reviewer mentioned, the products in biomass conversion are quite complex, including gaseous, liquid and solid phases. In the present work, we are mainly focusing on the liquid hydrocarbons, in which they are the main component of bio-oil or gasoline. So, the yield of liquid phase product was mainly discussed in the manuscript.

Gaseous products were collected and analyzed by gas-chromatography with flame ionization detector (GC-FID). However, due to the high initiate H₂ pressure, any gaseous products formed were highly diluted in the hydrogen gas, rendering accurate quantification challenging. Only trace amounts of methane, resulting from minor C-C bond cleavage in lignin side chains were detected, corresponding to a mass yield below the micromole level per gram of lignin. Given the negligible quantity, gaseous products were excluded from the overall carbon and mass yield calculation. We anticipate that this omission has a minimal impact on the reported total liquid hydrocarbon yield. On the other hand, the target product in this work is the liquid hydrocarbons, the main component of gasoline.

Note that all products and external standard are cyclic hydrocarbons; the relative response factors for each species were assumed to be 1.0 (*J. Lercher et al., Science*

2023, 379, 807–811). The relative response factors for all the products detected are ranging from 0.96-1.02. Therefore, to make the calculation consistency we chose the relative response factor to be 1 in the last submission version.

Regarding the solid residue, the reaction mixture after reaction was separated into liquid and solid phases using a filtration membrane. The collected solid, comprising both the catalyst and residual carbonaceous materials were weighted together, and the net residue mass was determined by subtracting the mass of catalyst. As noted by the reviewer and supported by the literature, accurate quantification of all solid residues remains challenging due to inevitable handling losses, incomplete recovery, and potential deposition on reactor walls, etc. Consequently, mass loss is expected in the mass balance, which should be taken into account in the overall mass closure assessment.

According to the reviewer’s suggestion, in this revised version, we have modified the calculation based on the relative response factors, and there are only small changes in the numbers after the decimal point. The mass yields are updated in the revised manuscript, as shown in Table S4 in the supporting information on page 17, and Figure 2 in the manuscript on page 9.

The following sentences were added to the method part in the supporting information on page 4.

“The corresponding response factors were determined by analyzing mixtures of pure, commercially obtained liquid hydrocarbons and an internal standard with precisely known masses.”

The following table has also been updated on page 17 in the supporting information

Table S4. Summary of product yields of direct HDO of lignin over different catalysts.

Catalyst	Mass yield (wt%)								Total (%)
	Cycloalkanes				Aromatics				
	C ₆	C ₇	C ₈	C ₉	C ₆	C ₇	C ₈	C ₉	
Ru/CNF	26.5	12.6	8.0	2.0	–	–	–	–	49.1
Ru/CNF-700	7.3	2.8	5.2	–	–	–	–	0.8	16.1

Ru/CNF-1000	0.8	3.0	4.4	1.8	-	-	-	1.3	11.3
Ru/AC	0.7	2.4	1.6	0.8	3.8	2.5	1.9	0.5	14.2
Ru/Al ₂ O ₃	0.3	1.3	0.3	-	3.0	2.2	1.8	0.7	9.6

Reaction conditions: 0.1 g lignin, W_{cat}=0.1 g, 20 ml dodecane, 8 h, 5 MPa H₂ at 250 °C.

The following figures have been updated in the manuscript on page 9:

Figure R25. Catalytic performance of HDO of lignin over the various Ru catalysts.

Comment 3. Although it is widely acknowledged that the carbon balance of biomass conversion is challenging to calculate, the authors report only liquid yields without quantifying the formation of gaseous by-products (C1-C4) or solid cokes. The absence of these data creates a significant gap in the carbon balance, potentially compromising the accuracy of product selectivity. The authors should try their best to trace all carbon atoms wherever possible, at the very least providing mass yields for gaseous products and solid cokes.

Response: We sincerely thank the reviewer for raising this insightful point regarding the carbon balance. We fully agree that a comprehensive mass balance, encompassing gaseous and solid products, is important for accurately assessing the performance and selectivity of biomass conversion processes.

As addressed in our response to “# comment 2”, the gaseous products were analyzed by GC-FID, however, due to the high initial H₂ pressure (50 bar), any gaseous species formed (notably only trace amounts of methane from minor C-C cleavage) were highly

diluted and fell below reliable quantification limits (less than micro mol per gram of lignin). Consequently, gaseous products were excluded from the mass yield calculations, and the reported yields are based on quantified liquid phase products, since they are the main component of gasoline.

A detailed explanation, along with corresponding revisions to the manuscript, has been provided in the response to “# Comment 2”.

Comment 4. There is a massive pressure gap between the NAP-XPS characterization (0.2 mbar) and the actual reaction conditions (5 MPa). The surface state of Ru under 50 bar of H₂ might be significantly more reduced than what is observed at 0.2 mbar. The authors should discuss this limitation or provide evidence.

Response: We thank the reviewer for this thoughtful critique regarding the pressure gap between our NAP-XPS characterization (0.2 mbar) and the actual reaction conditions (5 MPa H₂). We fully acknowledge this as an important limitation and agree that the surface state of Ru under 50 bar H₂ may be significantly more reduced than what is observed at 0.2 mbar. We have revised the manuscript to explicitly discuss this limitation and its implications for interpreting our results.

Bridging the gap between idealized surface science studies and practical catalytic conditions remains a fundamental challenge in heterogeneous catalysis. Industrial catalytic reactions often proceed under high temperatures and pressure, yet the operando characterization tools capable of probing catalysts under true realistic conditions remain limited by equipment constraints. This is particularly true for complex multiphase systems like lignin HDO, where gas, liquid, and solid phases coexist under elevated pressure.

Despite this limitation, we employed NAP-XPS because it offers significant advantages over conventional ultrahigh vacuum XPS for this system. Ruthenium is known to oxidize readily upon exposure to air, making it difficult to assess its chemical state using standard XPS under high vacuum. NAP-XPS allow us to migrate oxidation during transfer and measurement, providing a more authentic representation of the catalysts surface chemistry after pretreatment. As reviewed extensively (*Chem. Rev.* **2019**, *119*, 6822-6905), NAP-XPS has proven valuable for probing surface chemistries

under conditions approaching those of thermal catalysis, even if full industrial conditions remain inaccessible.

Nevertheless, we recognize that 0.2 mbar is nearly four orders of magnitude below 5 MPa, and the catalyst surface under high-pressure H₂ is likely to be more reduced than our NAP-XPS measurements indicate. However, the systematic variation in oxidation state observed at different pretreatment temperatures under NAP-XPS conditions is expected to reflect the relative trend that persists under reaction conditions, even if the absolute values differ.

In the revised manuscript, we have added the following sentences to the discussion part on page 12:

“To gain a comprehensive understanding of the local structure of Ru and its interaction with oxygen species on the CNF support, near-ambient pressure XPS (NAP-XPS) was performed. The samples were reduced under H₂ to mimic the reducing atmosphere of the HDO process; while ensuring they were not exposed to air during transfer or measurement. Nevertheless, we recognize that 0.2 mbar is nearly four orders of magnitude below the 5 MPa H₂ pressure used in catalytic testing, and the catalyst surface under high pressure H₂ is likely to be more reduced than our NAP-XPS measurement indicate. However, the systematic variation in oxidation states observed at different pretreatment temperature under NAP-XPS conditions is expected to reflect the relative trend that persists under reaction conditions, even if the absolute values differ.”

Comment 5. Regarding the potential for practical application, the authors provide real-life biomass as examples. Can authors provide scale-up to grams or hundreds of grams to demonstrate the practicality of this approach?

Response: We appreciate the reviewer’s kind suggestion. In the manuscript, Ru/CNF catalyst shows both high catalytic activity toward lignin and real beechwood biomass (Dansk Traemel) to liquid hydrocarbons via hydrodeoxygenation reaction in a one-pot process. It shows very high significance in the field of biomass conversion and utilization. We agree with the reviewer that it is very meaningful to perform the scale-up experiments.

In the revised version, we added the experiment with scaling up the feedstock of lignin-rich corncob residue to 0.5 g, the results shows that similar yield can be obtained over the Ru/CNF catalyst.

The following Figure R26 are added as Figure S15 on page 19 of the supporting information, and the following sentences has added in line 223-227 of the manuscript:

“To further assess the scalability of the one-pot conversion process from lignin to liquid hydrocarbons, a scale-up experiment was performed by increasing the feedstock amount fivefold, as shown in Figure S15. A total mass yield 44% was obtained, with cycloalkanes as the main products and no residual oxygenates detected, highlighting the promising scalability of the process.”

Figure R26. Catalytic performance of HDO of lignin over Ru/CNF catalyst. Reaction conditions: 0.5 g biomass, 0.5 g catalyst, 20 ml dodecane, 8 hours reaction, 10 MPa H₂ at 250 °C.

Due to the instrument limitations, further scale-up of the feedstock loading beyond the current level was not feasible.

The following sentences have been added to the manuscript on page 10.

“To further evaluate the applicability of the one-pot conversion of lignin to liquid hydrocarbons, the scale-up experiment was conducted by increasing the feedstock amount to five times higher, as shown in Figure S11. A total mass yield 44% was obtained, with cycloalkanes as the main products and no oxygenated compounds

detected. It demonstrates promising scalable potential for producing liquid hydrocarbons from lignin.”

Comment 6. The caption for Fig. S8 references "Scheme 3," but this scheme is not present in the manuscript or supplementary information. This seems to be a typo referring either to Fig. S8 itself or a figure in the main manuscript. Please verify and correct this citation to ensure consistency.

Response: we appreciate the reviewer’s careful check. In the revised version of the supporting information, we have corrected the typo error on page 16 in the supporting information:

Figure R27. The concept of cycloalkane as the benzene and H₂ carrier in the HDO process.

“This research not only provides a significant advancement in the one-pot HDO of lignin to hydrocarbons but also offers a promising approach for integrated hydrogen storage and the production of green aromatics through the selective conversion of lignin with green hydrogen to cycloalkanes, as shown in Figure S10. The use of cycloalkanes as a hydrogen carrier further highlights the potential of this process in facilitating the transportation and on-site release of hydrogen, underscoring the environmental and commercial viability of converting lignin, a byproduct of the biorefinery industry, into valuable products using Ru/CNF catalysts.”

We sincerely thank the reviewers for their careful review of our manuscript and for their valuable comments and suggestions, which have certainly helped us to improve our work.

Reviewer #1 (Remarks to the Author):

The authors have taken into account all my comments and to my opinion those of other reviewers.

Their manuscript is now much more robust than in their first submission. I have just now one minor revision query:

Although I appreciate the inclusion of Table S3 to position the Ru/CNF system within the current literature, the comparison remains heterogeneous and somewhat descriptive rather than rigorously quantitative. The listed studies involve different lignin types, solvents, pressures, temperatures, and product definitions, making direct comparison difficult.

Given that the manuscript claims outstanding performance, I recommend that the authors either (i) normalize the comparison where possible (e.g., clearly distinguish mass yield vs hydrocarbon yield, specify whether oxygenates are included, clarify feedstock type and severity of conditions), or (ii) more explicitly discuss the limitations of cross-study comparison and moderate any statements implying record-level performance.

Addressing this point would significantly strengthen the credibility and positioning of the work.

Response: Thank you so much for your kind words and for supporting the publication of our manuscript.

in the revised version, we have expanded the information to provide a clearer overview of the comparison with literature. Since our work focuses on converting lignin to liquid hydrocarbons, specifically cycloalkanes and aromatics, the table now strictly compares lignin conversion to hydrocarbons (C_xH_y), excluding oxygenated products.

Besides, we have moderated the statement in the manuscript regarding the mass yield obtained in this work. The following sentences has been modified on page 7 (line 157-159).

“As summarized in Table S3, the hydrocarbon yield reported here represents one of the highest values achieved to date for converting lignin or woody biomass into liquid hydrocarbons.”

Table S3. Summary of the converting lignin to **liquid hydrocarbons** from lignin and woody biomass. (TW: this work).

Catalyst	T (°C)	P (MPa)	Feedstock	Main products	Mass yield	Main products distribution	Ref.
Ir-ReO _x /SiO ₂	260	4	Organosolv lignin	Cycloalkanes	19.3%	Cyclohexane: 3.2 Methyl cyclohexane: 6.7 Ethyl cyclohexane: 2.2 Propyl cyclohexane: 7.2	11
			Enzymolysis lignin		9.8%	Cyclohexane: 3.6 Methyl cyclohexane: 0.7 Ethyl cyclohexane: 3.6 Propyl cyclohexane: 1.9	
			Alkaline lignin		6.6%	Cyclohexane: 3.1 Methyl cyclohexane: 0.5 Ethyl cyclohexane: 2.2 Propyl cyclohexane: 0.8	
Pt/NbOPO ₄	190	5	Birchwood	Pentanes, hexanes, alkylcyclohexanes	28.1%	Pentanes: 10.2 Hexanes: 13.1 Alkylcyclohexanes: 4.8	12
Pd/m-MoO ₃ -P ₂ O ₅ /SiO ₂	180	1	Wood and bark-derived-bio-oil	Pentane, hexane, methylcyclopentane, cyclohexanes	9.4%	Pentane: 0.9 Hexane: 1.2 Methylcyclopentane: 0.9 Cyclohexanes: 5.6	13
	250	1			29.6%	Pentane: 7.4 Hexane: 5.1 Methylcyclopentane: 3.7 Cyclohexanes: 13.4	
Ru/Nb ₂ O ₅	250	0.7	Birch lignin	Arenes, cyclohexanes	29.7%	Toluene: 2.8 Ethyl benzene: 9.1 Propyl benzene: 8.5 Methyl cyclohexane: 0.6 Ethyl cyclohexane: 4.4 Propyl cyclohexane: 3.6	14
Ni/SiO ₂ -Al ₂ O ₃	300	6	Corncob lignin	Alkanes	42%	Alkanes: 42	15
NiAl alloy	220	2	Poplar wood sawdust	Aromatic monomers	18.9%	Aromatic monomer: 18.9	16
Ru/Nb ₂ O ₅ -SiO ₂	230	-	Birch lignin	Arenes	19.8%	Toluene: 1.8 Ethyl benzene: 8.0 Propyl benzene: 5.0 arene dimers: 3.8 Cycloalkanes: 1.2	17
Ni-Cu/H-Beta zeolite	330	-	Kraft lignin	Cycloalkanes	40.39%	Cycloalkanes: 40.39	18
Ru/CNF	250	5 (at 250 °C)	Corncob	Cycloalkanes	49.1%	Cycloalkanes (C ₆ : 26.5, C ₇ : 12.6, C ₈ : 8.0, C ₉ : 2.0)	TW

Note: the pressures were reported as the initial pressure at room temperature in the literature unless specified.

Reviewer #2 (Remarks to the Author):

The author well addressed all my concerns. It can be published as is.

Response: Thank you for your review and for your recommendation to publish our manuscript.

Reviewer #3 (Remarks to the Author):

The authors properly responded to my comments and improved the manuscript in the revised manuscript. I recommend the acceptance of the manuscript for publication.

Response: Thank you very much for your positive feedback and for taking the time to review our manuscript.